# Socio-Demographic Correlates of Total and Domain-Specific Sedentary Behavior in Latin America: A Population-Based Study

**DOI:** 10.3390/ijerph17155587

**Published:** 2020-08-03

**Authors:** Gerson Luis de Moraes Ferrari, André Oliveira Werneck, Danilo Rodrigues da Silva, Irina Kovalskys, Georgina Gómez, Attilio Rigotti, Lilia Yadira Cortés Sanabria, Martha Cecilia Yépez García, Rossina G. Pareja, Marianella Herrera-Cuenca, Ioná Zalcman Zimberg, Viviana Guajardo, Michael Pratt, Cristian Cofre Bolados, Rodrigo Fuentes Kloss, Scott Rollo, Mauro Fisberg

**Affiliations:** 1Laboratorio de Ciencias de la Actividad Física, el Deporte y la Salud, Facultad de Ciencias Médicas, Universidad de Santiago de Chile, USACH, Santiago 7500618, Chile; cristian.cofre@usach.cl (C.C.B.); rfuentesk@gmail.com (R.F.K.); 2Department of Nutrition, School of Public Health, Universidade de São Paulo (USP), São Paulo 01246-904, Brazil; andreowerneck@gmail.com; 3Department of Physical Education, Federal University of Sergipe–UFS, São Cristóvão 49100-000, Brazil; danilorpsilva@gmail.com; 4Carrera de Nutrición, Facultad de Ciencias Médicas, Pontificia Universidad Católica Argentina, Buenos Aires C1107 AAZ, Argentina; ikovalskys@gmail.com (I.K.); viviana.guajardo@comunidad.ub.edu.ar (V.G.); 5Departamento de Bioquímica, Escuela de Medicina, Universidad de Costa Rica, San José 11501-2060, Costa Rica; georgina.gomez@ucr.ac.cr; 6Centro de Nutrición Molecular y Enfermedades Crónicas, Departamento de Nutrición, Diabetes y Metabolismo, Escuela de Medicina, Pontificia Universidad Católica, Santiago 833-0024, Chile; arigotti@med.puc.cl; 7Departamento de Nutrición y Bioquímica, Pontificia Universidad Javeriana, Bogotá 110231, Colombia; ycortes@javeriana.edu.co; 8Colégio de Ciencias de la Salud, Universidad San Francisco de Quito, Quito 17-1200-841, Ecuador; myepez@usfq.edu.ec; 9Instituto de Investigación Nutricional, La Molina, Lima 15026, Peru; rpareja@iin.sld.pe; 10Centro de Estudios del Desarrollo, Universidad Central de Venezuela (CENDES-UCV)/Fundación Bengoa, Caracas 1053, Venezuela; manyma@gmail.com; 11Departamento de Psicobiologia, Universidade Federal de São Paulo, São Paulo 04023-062, Brazil; iona.zimberg@gmail.com; 12Institute for Public Health, University of California San Diego, La Jolla, CA 92093-0021, USA; mipratt@health.ucsd.edu; 13Healthy Active Living and Obesity (HALO) Research Group, Children’s Hospital of Eastern Ontario Research Institute, Ottawa, ON K1H 8L1, Canada; arollo@cheo.on.ca; 14Department of Pediatrics, Faculty of Medicine, University of Ottawa, Ottawa, ON K1H 8M5, Canada; 15Instituto Pensi, Fundação José Luiz Egydio Setubal, Hospital Infantil Sabará, São Paulo 01227-200, Brazil; mauro.fisberg@gmail.com; 16Departamento de Pediatria da Universidade Federal de São Paulo, São Paulo 04023-061, Brazil

**Keywords:** sedentary behavior, screen-time, sitting, epidemiology, sociodemographic, Latin America

## Abstract

Purpose: The aim of this study was to identify socio-demographic correlates of total and domain-specific sedentary behavior (SB). Methods: Cross-sectional findings are based on 9218 participants (15–65 years) from the Latin American Study of Nutrition and Health. Data were collected between September 2014 and February 2015. Participants reported time spent in SB across specific domains. Sex, age, ethnicity, socioeconomic (SEL), and education level were used as sociodemographic indicators. Results: Participants spent a total of 373.3 min/day engaged in total SB. Men, younger adults, other ethnicities, higher SEL and educational level presented higher total SB when compared with women, older adults, white/Caucasian, and low SEL and educational level. Men spent more time on the playing videogames (*b*: 32.8: 95% CI: 14.6;51.1) and riding in an automobile (40.5: 31.3; 49.8). Computer time, reading, socializing or listening to music was higher in younger participants (<30 years) compared with those ≥50 years in the total sample. Compared to the low SEL and educational level groups, middle (11.7: 5.7; 17.6) and higher (15.1: 5.3; 24.9) SEL groups as well as middle (9.8: 3.6; 15.9) and higher (16.6: 6.5; 26.8) education level groups reported more time spent reading. Conclusion: Socio-demographic characteristics are associated with SB patterns (total and specific) across Latin American countries.

## 1. Introduction

In recent years, Latin America has experienced positive changes in public transport systems, increased female employment, rapid urbanization, industrial production patterns, and improved socioeconomic levels, leading to a decrease in the energy required to meet the burden of daily living [1]. Over the past few decades, health promotion and public health research incentives have focused mostly on physical exercise and physical activity, due to its established relationship with chronic disease risk reduction [2]. Unfortunately, people spend less than five percent of their time engaged in moderate-to-vigorous physical activity daily [3]. The majority of time is often spent in light physical activity or sedentary behavior (SB).

SB includes several different activities during waking hours, in which an individual is sitting, reclining, or lying down and requires low energy expenditure (≤1.5 metabolic equivalents) [4]. Sedentary time is associated with a greater risk for several major chronic disease outcomes, as well as cardiovascular and all-cause mortality [5,6,7]. To date, the majority of population-based evidence has been derived from studies using self-report exposure measures, typically with single item questions on television (TV) time, total sitting time [8], or time spent using print, broadcast, online, and social media [9,10]. The inclusion of such questions in epidemiological studies has provided informative insights into the prevalence of SB across different countries. For example, cross-country comparisons have reported wide variations in sitting time; countries such as Spain, and Northern Ireland, report 240.5 min/day in contrast to reports of 360.5 min/day in Sweden and Denmark [11]. In Latin American countries, variation in levels of sitting time have also been found across countries, ranging from 300 min/day in Ecuador to 480 min/day in Argentina and Peru [12]. Research in low, middle, and high-income countries has shown that large amounts of time during waking hours are spent being sedentary, specifically in sitting time [11,12]. Within countries, total and domain-specific sitting time has been shown to vary by indices of socio-demographic factors. For example, younger individuals (21–30 years-old) spend more time in leisure sitting time than older individuals (>61 years-old) [13]. Further, greater access to social media at home has been related to increased screen-based SB [14] and an elevated risk of developing poor mental health outcomes [15]. Such variations in domain-specific (watching TV, computer use at home, and riding in an automobile) sedentary time may have implications for health [16,17].

In light of this growing scientific interest, research is needed to examine the prevalence, spatial variation and sociodemographic correlates of SB. To date, published international comparisons have often operationalized SB using a single indicator for sedentary time (i.e., time spent sitting per day) and these results may be overestimated [18]. Previous findings have also indicated that associations between different domains of SB and health outcomes vary, especially for mental health [19,20,21,22]. In this sense, it has been shown that passive SB (e.g., watching TV, listening to music) were associated with overweight/obesity and elevated depressive symptoms, while mentally-active sedentary behaviors (e.g., office work) were not associated with risk of being overweight and favorably associated with depressive symptoms [20,21]. Similarly, previous findings have shown that sitting time related to transportation and watching TV may be more harmful for cardiovascular risk factors, when compared with occupational sitting time [19].

Therefore, identifying the duration of sitting time across different activities and differences between specific domains is important for informing future research, public health interventions, policies and practices for occupational health, urban city planning, and transportation initiatives [4]. To date, there is limited evidence on the associations of socio-demographic characteristics with total and domain-specific SB in low-middle income countries, including those in Latin America. Previous studies considering total sitting time found that young adult participants with higher educational levels presented higher overall sitting time [12,18]. However, correlates such as education and age may also be associated with time spent in different types of SB [23]. The aim of this study was to examine the associations between socio-demographic characteristics and total and domain-specific SB in a sample of adolescents and adults from eight Latin American countries.

## 2. Material and Methods

### 2.1. Study Design and Sample

The Latin American Study of Nutrition and Health (Estudio Latinoamericano de Nutrición y Salud; ELANS) is a cross-sectional, epidemiological, multi-national survey including eight Latin American countries (Argentina, Brazil, Chile, Colombia, Costa Rica, Ecuador, Peru, and Venezuela). Only data for urban locations were included to increase comparability across countries and for reasons surrounding feasibility [24].

Data collection occurred between September 2014 and February 2015. All investigators completed training and all countries met local ethics requirements. The ELANS protocol is registered at ClinicalTrials.gov (#NCT02226627) and was approved by the Western Institutional Review Board (#20140605). All participants signed the informed consent.

We considered a *p* < 0.05, a limit error of 3.5%, and a survey design effect of 1.75 for sample size calculation. The study was conducted with a complex and multi-stage cluster-stratified sample design, with all regions for each country represented and random selection of main cities in each region, according to the probability proportional to size. The sample was stratified by sex, age range and income level. Households within each secondary sampling unit were selected based on systematic randomization. Participants (15–65 years) were recruited and the final sample included 9218 (4409 [48.1%] men) adolescents and adults. Details have been previously published [24].

### 2.2. Self-Administered Total and Domain-Specific Sedentary Behavior

Total and domain-specific SB were self-reported using a questionnaire validated by Salmon et al. [25]. We adapted the questionnaire and asked the questions in the last week (last 7 days) and not separated by weekday and weekend. Participants were instructed to report the mean time spent in each behavior. For ELANS study, the questionnaire was translated into Spanish in accordance with the World Health Organization process of instrument translation and adaptation [26]. The original measure of the questionnaire demonstrated acceptable test-retest reliability for computer use at home (intraclass correlation coefficient [ICC]: 0.62; 95% CI: 0.48; 0.73), reading (books/magazines; ICC: 0.78; 95% CI: 0.69; 0.84), socializing with friends or family or listen to music/CD/radio (ICC: 0.76; 95% CI: 0.66; 0.82), talking on the telephone (ICC: 0.06; 95% CI: −0.13; 0.19), watching TV at home (ICC: 0.82; 95% CI: 0.75; 0.87), and time spent inside a motor vehicle (ICC: 0.85; 95% CI: 0.79; 0.89) [25]. Compared to accelerometer (ActiGraph model GTIM) data, the questionnaire has been shown to have good reliability (ICC: 0.52; 95% CI: 0.27; 0.70) and modest validity (ICC: 0.30; 95% CI: 0.02; 0.54), and be suitable for use in adults [27]. In addition, this questionnaire has been used in previously published studies [28,29].

The questionnaire asked about time spent in SB during a typical week across seven specific domains: (a) Computer use at home; (b) videogame use; (c) reading (books/magazines); (d) socializing with friends or family or listening to music/CD/radio; (e) talking on the telephone; (f) watching TV at home; and (g) time spent inside a motor vehicle (car, motorcycle, train or buses). Two questions were asked: (i) *“How many days did you use the computer at home in the last 7 days?”*; (ii) *“On average, how many minutes did it take you to use the computer at home on the days previously mentioned by you?”*. These questions were asked separately for all seven specific domains. Total time spent in SB was calculated as the sum of daily time spent sedentary in each specific domain and reported as min/day.

### 2.3. Sociodemographic Characteristics

Respondents self-reported age (15–65 years) and were categorized into three age groups (<30, 30–49, and ≥50 years) in order to obtain appropriate sample sizes. Sex, socioeconomic and educational level were classified using the same questionnaire in all countries [26,27,28,29,30,31]. As countries had different categories of socioeconomic level (SEL), a single three-tier system (low, medium, high) was developed. We carried out a similar process to standardize the level of education into three levels (basic or lower (low), high school (medium) and university degree (high)) in all countries [12,24,30]. Participants were asked about their ethnicity (white/Caucasian, black, mixed, other: Asian, indigenous, gypsy and other).

### 2.4. Statistical Analysis

The participants who provided complete information for the variables were analyzed in the present study. The Kolmogorov Smirnov test was applied to check the distribution of the data. Since not all SB domains were normally distributed, the descriptive statistics were presented as mean, median and 95% confidence intervals (95% CI). Furthermore, frequencies, percentages, values for the 25th and 75th percentile were also reported. For categorical analyses, we applied Chi-square tests. We also divided the total SB score at the median point (275 min/day) for categorical analysis because there is currently not an international recommendation available. It has to be noted that there is no consensus on the cut-off for sitting too much [4,8,31,32]. Results were stratified by country, sex, age group, ethnicity, and socioeconomic and educational levels.

Multilevel linear regression models, including region and cities as random effects, adjusted for sex, age, ethnicity, SEL, and education level, reporting unstandardized beta coefficients and 95% confidence intervals, were used to examine the associations between sociodemographic characteristics with each domain of SB, for each country and overall. All analyses were performed with SPSS V22 software (SPSS Inc., IBM Corp., Armonk, New York, NY, USA) [33]. The samples were weighted considering sociodemographic characteristics, sex, and socioeconomic level, to make the sample comparable with the whole population of each country [24].

## 3. Results

Descriptive characteristics of participants are presented in Table 1. Overall, 9218 participants aged ≥15 years (mean: 35.8, 95% CI: 35.5; 36.1) completed the questionnaire. For the total sample, mean and median total SB were 6.2 and 4.6 h/day, respectively. The proportion of participants who reported >275 min of total SB per day was greater than half. Table 1 shows the significant differences in total SB time between countries, and according to sex, age group, ethnicity, socioeconomic and education levels, as well as the proportion of the sample with a total SB of >275 min/day (Table 1).

The association between correlates and total SB is presented on Table 2. Men (60.9 min/day higher), younger adults (130.9 min/day higher), other ethnicities (89.5 min/day higher), higher SEL (119.2 min/day higher) and higher educational level (88.8 min/day higher) presented higher total SB when compared with women, older adults, white/Caucasian, low SEL and low educational level respectively.

In absolute terms, the highest values of domain-specific SB were reported for watching TV (146.3 min/day; 95% CI: 142.8; 149.8). The highest mean level of watching TV was observed in Costa Rica (220.3 min/day; 95% CI: 201.2; 239.4) followed by Brazil (172.3 min/day; 162.7; 184.3) (Appendix A). For the total sample, men showed greater SB than women on computer use at home (Peru), videogame use (Argentina, and Costa Rica), reading (Brazil), and riding in an automobile (all countries), regardless of age, ethnicity, SEL, and education level. Talking on the telephone was the only domain where there were no differences between sexes in all countries. On the other hand, men presented about 33 min/day higher videogame use and 41 min/day higher sedentary time riding an automobile than women (Table 3).

Computer time was higher in younger participants (<30 years) compared with those ≥50 years in Argentina, Brazil, Chile, Colombia and Ecuador regardless of sex, ethnicity, SEL and education level. Regarding videogame use, no differences between age groups were observed within each country and in total sample. Higher reading (all countries, except Brazil and Costa Rica) and socializing or listening to music (all countries, except Chile, Costa Rica and Peru) were observed among participants aged <30 years, compared to the older participants (≥50 years). Regarding watching TV, compared to the reference group (≥50 years), less time was observed among the middle-aged group (30–49 years) in Chile and more time was observed among the younger group (<30 years) in Ecuador. In the overall sample, younger adults reported 44.5 min/day higher computer use at home, 22.4 min/day higher reading, 20.8 min/day higher time spent listening to music and 15.2 min/day lower time riding in an automobile than older adults. No differences were observed between age group talking on the telephone and riding in an automobile in each country and in total sample (Table 4).

The association between ethnicity and domains of SB is presented in Table 5. There was a consistent variation across countries, with Brazil and Argentina presenting the largest differences, especially for other vs. white/Caucasian. In the overall sample, people of other ethnicities than white/Caucasion, black or mixed presented 44.5 min/day higher time spent in the computer at home, 47.7 min/day higher reading, 24.1 min/day higher time listening to music, 20.9 min/day higher time talking in the telephone and 40.1 min/day higher time watching TV than white/Caucasian participants.

No differences were observed between low vs. middle/high SEL groups with regards to computer use at home and socializing or listening to music (except in total countries). Compared to the low SEL group, higher (Brazil and Peru) SEL participants reported more time spent reading. Higher minutes of videogame use (Venezuela) were reported by the high SEL group compared to the low SEL group. In the overall sample, a higher SEL was only associated with a 15.1 min/day higher time spent reading (Table 6).

Greater time reading (all countries, except Costa Rica and Peru), talking on the telephone (Costa Rica) and riding in an automobile (Brazil) and using the computer at home (Brazil and Ecuador) was observed among those with a high education level. Also, the high education group reported less time watching TV compared to the low education group, especially in Chile and Peru. In the overall sample, a higher educational level was associated with 15.5 min/day higher time using the computer at home, 16.6 min/day higher time reading, 10.3 min/day higher time talking on the telephone as well as 14.8 min/day lower time watching TV when comparing with the lower educational level group (Table 7).

## 4. Discussion

The aim of this study was to examine the associations between socio-demographic characteristics and total and domain-specific SB in a sample of adolescents and adults from eight Latin American countries. With regard to total SB, men, younger adults, other ethnicities, higher SEL and higher educational level presented higher total SB when compared with women, older adults, white/Caucasian, low SEL and low educational level, respectively. Our main finding was that, although some patterns could be identified, the socio-demographic correlates of the specific domains of SB varied among Latin American countries. In general, men (computer use at home, videogame use, reading and riding in an automobile), younger individuals (<30 years) (computer use at home, videogame use, reading, socializing or listening to music), and those of high SEL (reading) and high educational levels (reading, talking on the telephone) were more sedentary. Further, individuals of other ethnicities than white/Caucasian, black or mixed presented higher time spent on the computer at home, reading, listening to music, talking on the telephone and watching TV than white/Caucasian participants. However, conflicting associations between countries were observed for some domains of SB, especially concerning different age, SEL and education levels.

In line with previous findings, higher SEL and education levels were correlated with greater total SB [34,35]. This may be due to those with higher SEL and education levels being more likely to be employed in more sedentary occupations. In our study the association between SEL and education level and total SB was independent of sex, age, and ethnicity. Most countries in the current study showed socioeconomic (Brazil, Chile, Ecuador, Peru, Venezuela and in all countries) and educational (Argentina, Brazil, Chile, Colombia, Ecuador, Venezuela and in all countries) gradients in total SB, with higher levels reported among higher SEL and education level groups. Presumably adults with higher education and from higher income groups have more sedentary jobs, are more likely to use cars versus active travel as a means of transport, and have more electronic entertainment and labor-saving devices at home. Studies have shown socioeconomic differences in the proportion of time spent in domain-specific SB; levels of TV viewing are higher among those in lower SEL, whereas occupational sitting time tends to be higher among those with higher educational attainment or income [23].

In 2018, the Global Action Plan on Physical Activity (2018–2030) adopted SB reduction as one of the plans for global chronic disease prevention and control [36]. The second edition of Physical Activity Guidelines for Americans also highlighted numerous knowledge gaps for making specific recommendations to reduce SB and its associated health outcomes [37]. In particular, understanding the landscape of SB is a critical step before population wide strategies can be developed and implemented. For instance, increases in domain-specific SB (e.g., leisure screen-time) have been presented in parts of high-income countries [38,39]. Descriptive and inferential epidemiological findings for specific domains of SB have not been widely reported, especially among low and middle income countries, such as those in Latin America. To our knowledge, our study was one of the first to examine socio-demographic correlates of specific domains of SB in this context.

In recent years, there has been a rapid accumulation of studies highlighting the distinct and harmful effects of SB on health outcomes. The 2018 United States Physical Activity Guidelines Advisory Committee (PAGAC) released a scientific report on SB and health, which found strong evidence of a dose-response relationship between both total SB and TV viewing and incident cardiovascular disease, as well as all-cause and cardiovascular disease mortality [40]. For instance, one meta-analysis reported that the risk of all-cause and cardiovascular disease mortality increased above a threshold of 6 to 8 h per day for total SB and 3 to 4 h per day for TV viewing [5]. Another study reported a dose-response relationship between daily sedentary time and the metabolic syndrome, characterized by an odds ratio of 1.09 (95% CI 1.01; 1.18) for each hour of SB [5,40,41]. Accordingly, more attention has been paid to the specific manifestation of this behavior. Excessive waking activities in a sitting, reclining or lying posture with an energy expenditure ≤1.5 metabolic equivalents (METs) [4] have been shown to be influenced by several different characteristics and associated with poor health outcomes [42,43]. There is insufficient evidence to verify the association between domain-specific SB (i.e., computer and videogame use at home, reading, socializing or listening to music, talking on the telephone and riding in an automobile) and health outcomes [44]. Given that domain-specific SB could potentially be a key factor in the relationship with health outcomes [42,44]. In this sense, studies about the frequency and distribution of SB across specific domains and/or activities could inform specific strategies for SB reduction and improve health status at the population level.

Among adolescents and adults from eight Latin American countries, we identified some general patterns regarding socio-demographic correlates of SB. It was found that Latin Americans spend between 60–150 min/day riding in an automobile and, for all countries studied, men reported more minutes riding in an automobile than women. This finding could potentially be explained by cultural norms, where driving is still more prevalent among men and jobs requiring driving are also generally occupied by men in these countries. In two countries, it was observed that men also spent more time using the computer at home (Brazil and Peru) and playing videogames (Argentina and Costa Rica), while positive associations (higher time among men) were observed for reading only in Brazil, and socializing or listening to music only in Chile. Watching TV was the only domain of SB where women reported greater levels than men (Peru). Taken together, specific strategies focusing on the reduction of SB among men and women should consider differences in levels of domain-specific SB according to sex, as well as how differences between sexes may vary across individual countries.

We also observed that the amount of time spent in each domain of SB varied between countries according to age groups. Younger groups (<30 years) were generally more sedentary than older groups (≥50 years), which do not corroborate previous findings [45]. This finding could be due to the procedures of data collection and instruments used. Several of the domains assessed in this study may have been more applicable to younger people, such as computer use at home and videogame use as well as listening to music. In addition, total SB may have been underestimated especially for older adults, in the domains of socializing, talking on the telephone, and watching TV. In most countries studied, we observed that the younger group (<30 years) reported more time reading and socializing or listening to music. This result was expected since younger groups are more likely to be at school and/or university where reading is a prerequisite. Further, especially with the wide access to smartphones and use of online social media platforms, younger people spend more time socializing via technology.

Overall, adjusted analyses showed that other ethnicities presented higher total SB when compared with white/Caucasian. In the overall sample, participants of other ethnicities than white/Caucasian, black or mixed presented higher time spent on the computer at home, reading, listening to music, talking on the telephone and watching TV than white/Caucasian participants. Our analyses revealed additional differences between ethnic groups in terms of influential socio-demographic and lifestyle factors of sitting that warrant attention. Culturally appropriate health promotion programs seem to be more effective than usual care or other control conditions [46]. The so-called “cultural targeting” of health promotion programs can be achieved in several ways, for example by providing project materials in participants’ native language or showing participants the impact of a certain health problem on their ethnic group [47]. In light of our findings, we suggest for the Latin American region context to consider other ethnicities as a separate target group when developing interventions aiming to reduce SB.

Few differences in time spent in each SB domain between socioeconomic groups were observed. The only consistent association was found for time spent reading, wherein higher socioeconomic level was associated with greater reading time. Similar trends for this SB domain were found for educational level, potentially due to these individuals spending greater time studying and being employed in more sedentary occupations (e.g., office work). Greater differences in reading time according to socioeconomic and educational level were found in Brazil, which could be interpreted as an indicator of social inequality and inform specific interventions. On the other hand, participants with higher educational levels reported less time spent watching TV (especially in Argentina, Chile, Costa Rica and Peru), which can also guide potential interventions among lower educational level groups, considering the harmful effect of watching TV for several health outcomes [5,21].

The focus of the present study and many others was on the factors that influence SB, which has previously been mainly on the individual level factors, such as biological, psychological, and behavioral [13,48]. However, these factors are not independent and addressing them in isolation will not result in a significant change in SB [48]. Social, environmental and political factors also need to be taken into account. A systematic review that examined SB correlates in adults identified numerous intrapersonal factors correlated to SB, many of which are not modifiable (e.g., age and ethnicity) [49]. However, they did not identify many factors or correlates outside the individual. Potentially significant factors, such as built, physical, social, and political environments, need to be identified. There are several studies that have investigated the environmental influences on SB, both at the individual level and in the community [23].

The utilization of subjective measurement is a good method for epidemiological population-based studies, and it is imperative that these measurements are as accurate and reliable as possible [29]. Self-report questions about domain-specific SB in a typical week are a useful metric that have been widely used in previous sedentary research [50,51]. Marshall et al. [52] compared total self-reported SB to day-specific accelerometer-based SB and reported a very low agreement, as shown by the Bland-Altman results. A systematic review by Helmerhorst et al. [53] reported that a median Spearman’s rho of 0.23 was typically found between self-report and accelerometry-derived SB. We showed that the correlation of sitting time estimates obtained using International Physical Activity Questionnaire (long version) and accelerometry was low [54]. The combination of large and relative underestimation and low precision is also likely to significantly reduce the ability to detect associations with outcomes [55,56]. This may explain publications that report different relationship with health outcomes between subjective and objectively evaluated SB. The accelerometer mounted on the waist may have limitations to be used as a reference method to detect sedentary activities (i.e., inability to differentiate between standing still and sitting down). In future examinations, it might be possible to use different objective tools that differentiate these movements more precisely. Our results contribute to the literature emphasizing the association of domain-specific SB evaluated by subjective methods with sociodemographic correlates.

To our knowledge, this was the first multinational study, with nationally representative samples from middle-income countries, to analyze the distribution and sociodemographic correlates of both total and domain-specific SB. Despite several strengths, the current study had limitations that should be considered. First, some manifestations of SB were not present in the instrument used, such as sitting time at work and time spent using smartphones, which may have contributed to an underestimation of total SB and/or overestimation of time spent in specific SB domains (e.g., smartphone use reported on the “socializing or listening to music” domain). In a large sample from Australia, self-reported SB in occupational domain-specific contexts showed small significant associations with cardiometabolic biomarkers [19]. On the other hand, Wijndaele et al. [57] showed poor psychometric properties, for the items determining the number of breaks in occupational sitting, indicating the difficulty of recalling this irregular behavior in a reliable and accurate manner. In addition, the reproducibility of the domain-specific SB item, talking on the telephone, was low (0.06) [25]. Future population-based surveillance studies investigating levels and correlates of both total and domain-specific SB in Latin American adolescents and adults should include measures of occupational SB and various screen-based SBs. Evidence-based public health strategies and health promotion interventions are needed to address sitting-related health risks in the occupational setting, as well as the health burden of physically inactive commuting, prolonged periods of time spent sitting in an automobile and watching TV among Latin American adolescents and adults. This focus is of particular relevance, given the pace of change not only in communication technology and the conditions of work but also more broadly in people’s conditions of life in Latin America [9].

## 5. Conclusions

The findings from this study demonstrate considerable variation in levels of total and domain-specific SB according to sex, age, and socioeconomic and education levels across eight Latin American countries. The different associations found between countries have implications for future research and may inform intervention development at the regional and national levels. Strategies and intervention studies are needed to reduce SB in different domains, but predominantly time spent watching TV, in Latin America, which can be dependent on sociodemographic correlates such as sex, age, socioeconomic, and education level.

## Figures and Tables

**Table 1 ijerph-17-05587-t001:** Characteristics of participants by sociodemographic variables and the comparison with time spent in total sedentary behavior.

Variables	N	%	Mean (95%CI) of Min/Day	Median (25–75) of Min/Day	*p* ^a^	Percentage of >275 Min/Day (95%CI)	*p* ^b^
Total	9218	100.0	373.3 (366.0; 381.5)	275.0 (165.0–445.0)		51.7 (50.6; 52.7)	
**Countries**					<0.0001		<0.0001
Argentina	1266	14.4	369.3 (354.9; 382.7)	310.0 (190.0; 480.0)		58.6 (55.9; 61.3)	
Brazil	2000	21.3	455.1 (430.8; 480.2)	315.0 (185.0; 520.0)		55.6 (53.6; 57.8)	
Chile	879	9.8	303.1 (289.7; 317.0)	250.0 (155.0; 395.0)		43.8 (40.5; 47.3)	
Colombia	1230	13.2	367.2 (349.3; 384.8)	300.0 (170.0; 480.0)		54.1 (51.4; 57.0)	
Costa Rica	798	9.0	524.6 (487.8; 563.1)	365.0 (215.0; 611.2)		61.8 (58.5; 65.2)	
Ecuador	800	8.5	261.5 (250.3; 273.4)	220.0 (136.2; 350.0)		37.3 (34.3; 41.2)	
Peru	1113	12.8	347.4 (331.9; 363.5)	290.0 (182.5; 450.0)		53.0 (50.2; 55.9)	
Venezuela	1132	10.9	292.4 (281.4; 304.0)	250.0 (170.0; 370.0)		42.2 (39.1; 45.2)	
**Sex**					<0.0001		<0.0001
Men	4409	48.1	405.1 (393.7; 418.1)	310.0 (190.0; 490.0)		56.9 (55.4; 58.4)	
Women	4809	51.9	344.1 (336.0; 353.7)	260.0 (160.0; 422.5)		46.9 (45.5; 48.5)	
**Age group**					<0.0001		<0.0001
<30 years	3632	39.4	433.8 (418.9; 447.9)	334.5 (210.0; 515.0)		61.0 (59.4; 62.7)	
30–49 years	3696	55.3	349.9 (339.7; 361.3)	270.0 (170.0; 425.0)		48.7 (47.2; 50.2)	
≥50 years	1890	5.3	302.8 (290.2; 317.7)	235.0 (135.0; 375.0)		39.5 (37.3; 41.5)	
**Ethnicity**					0.278		0.266
White/Caucasian	3378	36.6	372.1 (359.7; 383.9)	290.0 (277.5; 300.0)		51.7 (50.0; 53.5)	
Black	593	6.4	396.4 (366.0; 429.2)	310.0 (288.0; 330.0)		55.1 (51.1; 59.4)	
Mixed	4078	44.3	355.2 (346.1; 365.7)	285.0 (275.0; 290.0)		51.1 (49.6; 52.6)	
Other	1169	12.7	461.7 (415.2; 507.1)	300.0 (273.0; 320.0)		53.1 (49.6; 56.8)	
**Socioeconomic level**					<0.0001		<0.0001
Low	4796	51.6	342.0 (332.5; 352.7)	255.0 (155.0; 420.0)		46.1 (44.6; 47.5)	
Middle	3542	39.0	393.8 (381.3; 405.8)	309.0 (190.0; 480.0)		55.9 (54.2; 57.6)	
High	880	9.4	461.2 (433.7; 494.8)	360.0 (235.0; 540.0)		65.2 (62.0; 68.4)	
**Education level**					<0.0001		<0.0001
Low	5643	61.2	346.9 (336.6; 356.9)	260.0 (155.0; 420.0)		46.6 (45.3; 47.9)	
Middle	2697	29.5	408.1 (394.5; 423.7)	320.0 (200.0; 495.0)		58.5 (56.6; 60.4)	
High	878	9.3	435.3 (365.4; 381.7)	350.0 (225.0; 500.0)		63.3 (60.0; 66.5)	

^a^ Mann-Whitney or kruskal-Wallis test for the comparison of medians; ^b^ chi-square for heterogeneity.

**Table 2 ijerph-17-05587-t002:** Adjusted analyses (*b* coefficient (95% CI)) between independent variables and total sedentary behavior by country.

Independent Variables	Argentina	Brazil	Chile	Colombia	Costa Rica	Ecuador	Peru	Venezuela	Overall
Sex ^1^									
Men vs. Women	32.7 (5.2; 60.3)	120.9 (72.4; 169.4)	30.1 (2.8; 57.5)	50.5 (16.2; 84.8)	81.1 (2.9; 159.3)	58.3 (33.6; 82.9)	52.3 (21.9; 82.8)	25.1 (2.5; 47.7)	60.9 (45.7; 76.3)
Age group ^2^									
30–49 years vs. ≥50 years	45.9 (11.6; 80.1)	54.4 (6.4; 102.5)	17.4 (−14.1; 48.9)	85.4 (48.0; 122.7)	52.2 (−53.9; 158.4)	59.2 (24.3; 94.1)	27.8 (−11.8; 67.6)	14.5 (−14.8; 43.7)	47.2 (29.6; 64.8)
<30 years vs. ≥50 years	100.6 (64.7; 136.6)	221.6 (143.6; 299.7)	120.9 (84.8; 157.0)	177.8 (133.2; 222.4)	85.3 (−23.0; 194.7)	103.4 (70.0; 136.8)	110.1 (66.3; 153.9)	74.2 (42.2; 106.3)	130.9 (108.7; 152.3)
Ethnicity ^3^									
Black vs. white/Caucasian	−36.5 (−317.3; 244.4)	26.8 (−20.2; 73.7)	16.9 (−10.5; 37.4)	25.1 (−45.4; 95.6)	−140.3 (−463.7; 183.1)	17.8 (−84.9; 120.6)	0.5 (−141.0; 142.0)	−15.4 (−67.4; 36.5)	24.3 (−6.9; 55.4)
Mixed vs. white/Caucasian	−1.1 (−33.9; 31.8)	99.5 (40.5; 158.5)	57.6 (26.5; 88.6)	−7.4 (−46.3; 31.4)	−33.6 (−127.2; 59.9)	−11.8 (−59.9; 36.5)	20.4 (25.9; 66.7)	30.0 (5.5; 54.5)	−16.9 (−32.4; −1.6)
Other vs. white/Caucasian	68.1 (−18.9; 155.2)	141.4 (74.6; 208.1)	22.5 (−48.4; 93.4)	64.6 (−26.9; 156.0)	−81.1 (−216.5; 54.3)	−93.4 (−182.9; −3.8)	−20.8 (−146.5; 104.8)	41.5 (−1.1; 84.1)	89.5 (56.1; 122.9)
Socioeconomic level ^4^									
Middle vs. low	18.4 (−9.8; 46.7)	23.8 (−23.9; 71.6)	42.3 (14.1; 70.5)	57.8 (19.7; 95.8)	30.6 (−56.2; 117.4)	56.4 (30.2; 82.5)	−7.1 (−40.9; 26.6)	24.9 (−3.7; 53.4)	51.8 (36.1; 67.5)
High vs. low	52.1 (−10.6; 114.7)	210.3 (114.9; 305.7)	99.2 (52.3; 146.1)	13.4 (−61.3; 87.9)	78.3 (−60.5; 217.2)	108.7 (72.3; 145.1)	88.1 (44.9; 131.3)	136.3 (87.5; 185.1)	119.2 (92.5; 145.8)
Education level ^5^									
Middle vs. low	89.0 (55.2; 122.9)	39.2 (9.0; 87.5)	60.8 (29.0; 92.6)	89.6 (49.3; 129.9)	91.5 (−28.0; 211.0)	106.7 (67.5; 145.9)	31.1 (−6.5; 68.8)	−4.4 (−34.9; 26.1)	61.1 (44.3; 77.9)
High vs. low	113.6 (47.5; 179.6)	210.1 (112.7; 307.6)	92.6 (49.5; 135.7)	73.5 (23.6; 123.4)	0.8 (−168.6; 167.1)	118.0 (69.4; 166.7)	5.5 (−47.7; 58.6)	103.8 (74.4; 133.3)	88.8 (62.4; 115.3)

Multilevel linear regression models, including region and cities as random effects: ^1^ Adjustment: age, ethnicity, socioeconomic and education level; ^2^ Adjustment: sex, ethnicity, socioeconomic and education level; ^3^ Adjustment: sex, age, socioeconomic and education level; ^4^ Adjustment: sex, age, ethnicity, and education level; ^5^ Adjustment: sex, age, ethnicity, and socioeconomic level; References: sex: women; age group: ≥50 years: ethnicity: white/Caucasian; socioeconomic level: low; education level: low; CI: confidence interval.

**Table 3 ijerph-17-05587-t003:** Adjusted analyses (*b* coefficient (95% CI)) between sex and sedentary behavior by country for specific-domains.

Country	Computer at Home	Videogame Use	Reading	Socializing or Listen to Music	Talking on the Telephone	Watching TV	Riding in an Automobile
Argentina							
Men vs. Women	−12.9 (−29.1; 3.2)	38.6 (7.1; 70.2)	3.8 (−8.0; 15.5)	−2.8 (−16.9; 10.9)	−0.4 (−8.0; 7.1)	−3.7 (−9.4; 16.8)	23.9 (2.9; 44.8)
Brazil							
Men vs. Women	27.5 (−1.1; 56.0)	12.3 (−62.7; 87.5)	24.8 (2.7; 47.0)	6.3 (−8.4; 21.1)	0.4 (−9.3; 9.5)	18.9 (−1.6; 39.4)	36.3 (22.5; 50.7)
Chile							
Men vs. Women	−7.5 (−25.7; 10.5)	15.9 (−21.2; 53.1)	3.6 (−7.7; 14.9)	11.7 (0.36; 23.1)	−1.1 (−9.9; 7.8)	−10.5 (−21.6; 0.50)	18.7 (3.5; 33.8)
Colombia							
Men vs. Women	5.2 (−17.4; 27.8)	28.4 (−3.5; 60.3)	5.8 (−4.6; 16.2)	−4.7 (−18.9; 9.6)	−4.5 (−15.0; 6.0)	−5.4 (−20.4; 9.4)	59.5 (29.7; 89.3)
Costa Rica							
Men vs. Women	8.3 (−43.2; 59.8)	99.7 (21.6; 178.0)	−8.1 (−39.6; 23.4)	10.9 (−15.3; 37.4)	−12.8 (−27.2; 1.5)	−17.6 (−56.8; 21.7)	89.8 (33.5; 146.1)
Ecuador							
Men vs. Women	3.4 (−10.7; 17.7)	13.4 (−14.2; 41.0)	1.8 (−10.4; 6.9)	5.5 (−4.3; 15.4)	0.8 (−6.2; 7.8)	2.1 (−7.4; 11.7)	21.7 (1.5; 41.9)
Peru							
Men vs. Women	31.05 (5.9; 56.2)	8.2 (−31.1; 47.6)	4.5 (−2.3; 11.2)	−5.1 (−16.0; 5.6)	1.7 (−7.2; 10.6)	−24.6 (−40.9; −8.4)	70.5 (20.1; 120.9)
Venezuela							
Men vs. Women	4.8 (−8.2; 17.8)	21.3 (−11.9; 54.5)	−4.3 (−11.6; 2.9)	−2.7 (−12.0; 6.5)	0.5 (−6.8; 7.8)	−4.8 (−13.8; 4.2)	31.2 (8.2; 54.1)
Overall							
Men vs. Women	8.1 (−1.1; 17.3)	32.8 (14.6; 51.1)	4.5 (−1.1; 10.1)	1.7 (−3.6; 6.9)	−1.6 (−5.1; 1.9)	−2.8 (−9.6; 4.1)	40.5 (31.3; 49.8)

Multilevel linear regression models, including region and cities as random effects, adjusted for age, ethnicity, socioeconomic and education level. Reference: women; CI: confidence interval.

**Table 4 ijerph-17-05587-t004:** Adjusted analyses (*b* coefficient (95% CI)) between age group and sedentary behavior by country for specific-domains.

Country	Computer at Home	Videogame Use	Reading	Socializing or Listen to Music	Talking on the Telephone	Watching TV	Riding in an Automobile
Argentina							
30–49 years vs. ≥50 years	6.4 (−16.2; 28.9)	11.9 (−42.3; 66.2)	−3.3 (−16.7; 9.9)	15.4 (−1.2; 31.9)	−6.4 (−15.6; 2.8)	−3.5 (−21.2; 14.2)	10.6 (−14.7; 36.1)
<30 years vs. ≥50 years	36.6 (11.6; 61.6)	−12.6 (−68.3; 43.1)	15.5 (1.7; 29.2)	30.2 (9.2; 51.1)	1.0 (−9.9; 11.9)	−1.9 (−19.8: 16.0)	4.1 (−27.8; 36.2)
Brazil							
30–49 years vs. ≥50 years	−7.9 (−43.8; 27.9)	−1.3 (−103.7; 101.0)	−8.4 (−30.1; 13.3)	−2.8 (−19.1; 13.4)	12.1 (−0.8; 24.9)	−14.2 (−39.0; 10.6)	−4.1 (−24.0; 15.9)
<30 years vs. ≥50 years	65.7 (4.5; 127.1)	4.8 (−173.5; 183.1)	36.4 (−0.2; 73.1)	32.1 (6.5; 57.7)	10.0 (−2.2; 22.4)	5.8 (−24.4; 36.1)	−19.2 (−37.7; 0.6)
Chile							
30–49 years vs. ≥50 years	12.9 (−14.4; 40.2)	−3.5 (−57.3; 50.1)	1.8 (−6.9; 10.5)	−0.9 (−17.3; 15.4)	−2.3 (−16.6; 12.1)	−19.1 (−33.5; −2.7)	−2.7 (−28.8; 23.3)
<30 years vs. ≥50 years	61.1 (25.0; 97.1)	21.3 (−65.6; 108.3)	32.6 (15.7; 49.5)	17.6 (−0.8; 35.9)	−4.8 (−16.4; 6.7)	−5.1 (−22.7; 12.5)	−16.9 (−35.6; 1.7)
Colombia							
30–49 years vs. ≥50 years	23.8 (−15.6; 63.2)	23.7 (−19.9; 67.2)	8.8 (−3.4; 21.0)	1.9 (−15.9; 19.8)	3.5 (−9.9; 17.1)	−0.6 (−18.9; 17.8)	−10.9 (−58.4; 36.5)
<30 years vs. ≥50 years	58.3 (16.8; 99.7)	63.7 (−10.2; 137.7)	21.6 (6.9; 36.4)	22.3 (2.7; 41.8)	10.8 (−3.2; 24.9)	9.1 (−12.9; 31.2)	−24.2 (−67.8; 19.3)
Costa Rica							
30–49 years vs. ≥50 years	−14.9 (−117.6; 87.6)	56.4 (−163.0; 275.9)	17.9 (−29.3; 65.1)	−2.6 (−41.6; 36.4)	11.5 (−9.3; 32.5)	−41.8 (−97.8; 14.3)	58.9 (−34.3; 152.3)
<30 years vs. ≥50 years	−25.6 (−124.4; 73.2)	67.8 (−84.3; 219.9)	28.3 (−7.6; 64.3)	16.2 (−26.6; 59.0)	14.4 (−5.0; 33.8)	−34.4 (−92.9; 24.2)	−29.7 (−96.9; 37.6)
Ecuador							
30–49 years vs. ≥50 years	28.0 (−2.5; 58.6)	6.7 (−30.2; 43.8)	5.7 (−4.0; 15.4)	7.4 (−7.2; 22.0)	1.3 (−9.9; 12.5)	11.9 (−1.6; 25.5)	12.7 (−21.3; 46.6)
<30 years vs. ≥50 years	45.5 (17.2; 73.7)	45.9 (−36.8; 128.8)	15.4 (3.1; 27.7)	18.1 (3.5; 32.7)	−0.6 (−10.0; 8.8)	16.4 (2.9; 29.9)	−8.0 (−35.7; 19.8)
Peru							
30–49 years vs. ≥50 years	12.6 (−21.9; 47.2)	28.9 (−42.2; 100.1)	0.7 (−7.3; 7.5)	−3.9 (−20.1; 12.2)	−3.4 (−16.0; 9.2)	17.9 (−5.6; 41.6)	−13.1 (−95.0; 68.9)
<30 years vs. ≥50 years	40.4 (−23.3; 104.1)	59.4 (−47.2; 166.1)	20.5 (10.2; 30.9)	5.7 (−9.9; 21.5)	−7.1 (−20.8; 6.5)	20.8 (−2.6; 44.1)	−49.5 (−112.5; 13.4)
Venezuela							
30–49 years vs. ≥50 years	−10.8 (−33.3; 11.7)	20.8 (−18.9; 60.5)	0.6 (−5.5; 6.8)	0.05 (−12.9; 13.1)	1.0 (−8.8; 10.8)	−7.8 (−20.5; 4.9)	5.0 (−31.4; 41.5)
<30 years vs. ≥50 years	5.5 (−203.; 31.2)	36.8 (−29.6; 103.2)	14.0 (2.8; 25.4)	14.9 (0.4; 29.4)	8.6 (−2.5; 19.8)	−5.7 (−18.9; 7.5)	−15.9 (−47.2; 15.3)
Overall							
30–49 years vs. ≥50 years	5.8 (−7.8; 19.6)	22.3 (−20.4; 55.8)	2.1 (−4.3; 8.6)	2.7 (−4.0; 9.5)	1.8 (−3.0; 6.7)	−5.2 (−14.4; 3.9)	8.0 (−6.4; 22.5)
<30 years vs. ≥50 years	44.5 (26.4; 62.5)	35.2 (−1.4; 71.9)	22.4 (14.3; 30.5)	20.8 (12.5; 29.2)	3.5 (−1.2; 8.3)	3.4 (−6.5; 13.4)	−15.2 (−27.4; −3.1)

Multilevel linear regression models, including region and cities as random effects, adjusted for sex, ethnicity, socioeconomic and education level. Reference: ≥50 years; CI: confidence interval.

**Table 5 ijerph-17-05587-t005:** Adjusted analyses (*b* coefficient (95% CI)) between ethnicity and sedentary behavior by country for specific-domains.

Country	Computer at Home	Videogame Use	Reading	Socializing or Listen to Music	Talking on the Telephone	Watching TV	Riding in an Automobile
Argentina							
Black vs. white/Caucasian	−73.9 (−284.9; 137.2)	−56.4 (−203.8; 90.9)	−2.3 (−13.5; 8.9)	0.9 (−229.8; 231.8)	−0.3 (−8.4; 8.9)	−25.7 (−227.1; 175.8)	22.5 (−19.2; 64.2)
Mixed vs. white/Caucasian	−4.4 (−26.7; 17.9)	−10.4 (−40.2; 19.4)	2.8 (−9.8; 15.5)	4.2 (−14.6; 23.0)	−6.3 (−15.9; 3.2)	0.5 (−17.0; 17.9)	19.2 (−12.8; 51.2)
Other vs. white/Caucasian	17.4 (−44.8; 79.6)	−31.6 (−138.1; 74.8)	56.8 (14.8; 98.8)	80.4 (24.1; 136.8)	26.4 (2.6; 50.2)	45.2 (3.3; 87.1)	−30.6 (−104.8; 43.5)
Brazil							
Black vs. white/Caucasian	31.3 (−22.6; 85.2)	−47.6 (−173.3; 78.1)	23.8 (−2.7; 50.3)	−5.6 (−29.5; 18.2)	−0.3 (−13.6; 13.1)	18.8 (−7.5; 45.1)	14.1 (−15.6; 43.8)
Mixed vs. white/Caucasian	31.8 (−25.9; 89.6)	38.8 (−137.1; 214.7)	33.9 (−0.5; 68.3)	41.2 (8.4; 74.1)	11.9 (−4.4; 28.3)	55.1 (20.2; 89.9)	−15.1 (−35.4; 5.2)
Other vs. white/Caucasian	73.5 (9.1; 137.9)	−31.2 (−175.6; 113.1)	108.2 (55.8; 160.6)	48.9 (16.8; 80.9)	33.3 (10.8; 55.8)	71.2 (35.2; 107.1)	23.0 (−6.1; 52.2)
Chile							
Black vs. white/Caucasian	17.9 (−36.6; 71.6)	−8.5 (−79.8; 62.9)	−4.6 (−15.5; 6.6)	6.1 (−19.3; 31.5)	5.9 (−19.0; 30.0)	1.3 (−32.5; 35.1)	3.5 (−35.4; 42.1)
Mixed vs. white/Caucasian	29.8 (5.5; 54.1)	13.5 (−18.6; 45.6)	4.3 (−6.6; 15.2)	7.8 (−8.3; 23.9)	4.8 (−8.4; 18.0)	9.5 (−5.1; 24.2)	−14.6 (−34.7; 5.5)
Other vs. white/Caucasian	23.2 (−28.5; 74.9)	142.6 (43.1; 242.2)	17.5 (−3.5; 38.5)	4.3 (−39.7; 48.3)	−20.4 (−47.4; 6.6)	−5.5 (−42.6; 31.7)	−20.7 (−78.3; 36.8)
Colombia							
Black vs. white/Caucasian	−12.2 (−58.5; 34.1)	24.8 (−40.1; 89.8)	1.1 (−18.9; 21.1)	−3.2 (−35.6; 29.3)	5.3 (−16.6; 27.1)	18.2 (−15.7; 52.2)	1.8 (−70.3; 73.9)
Mixed vs. white/Caucasian	2.1 (−30.8; 34.9)	4.0 (−34.1; 42.1)	12.1 (−1.9; 26.2)	−7.8 (−26.1; 10.3)	2.1 (−11.9; 16.1)	−9.7 (−27.3; 8.1)	16.8 (−26.3; 59.8)
Other vs. white/Caucasian	20.2 (−39.9; 80.4)	−42.6 (−100.9; 15.7)	10.4 (−15.8; 36.6)	29.3 (−9.4; 68.1)	35.1 (7.8; 62.5)	12.3 (−30.5; 55.1)	−31.0 (−114.5; 52.5)
Costa Rica							
Black vs. white/Caucasian	−2.9 (−245.6; 239.7)	−27.1 (−417.9; 363.7)	−13.9 (−126.9; 99.1)	−13.4 (−110.0; 83.4)	15.4 (−26.9; 57.7)	−90.1 (−234.9; 54.6)	−64.1 (−321.9; 193.6)
Mixed vs. white/Caucasian	−13.8 (−85.3; 57.8)	−35.8 (−137.2; 65.6)	10.8 (−32.6; 54.2)	−8.7 (−43.1; 25.7)	8.1 (−5.7; 21.9)	34.7 (−15.5; 84.9)	−80.0 (−153.7; −6.4)
Other vs. white/Caucasian	−49.2 (−145.5; 47.2)	−62.8 (−215.1; 89.4)	−13.9 (−70.5; 42.8)	−29.3 (−72.3; 13.7)	10.1 (−11.4; 31.7)	−10.2 (−74.2; 53.8)	−32.1 (−129.5; 65.4)
Ecuador							
Black vs. white/Caucasian	21.9 (−45.9; 89.7)	−0.8 (−932.7; 931.1)	10.7 (−23.5; 44.9)	4.3 (−38.4; 46.9)	54.4 (22.5; 86.3)	20.8 (−14.9; 56.5)	−2.5 (−85.7; 80.6)
Mixed vs. white/Caucasian	4.6 (−21.6; 30.8)	7.9 (−55.4; 71.5)	−1.9 (−16.0; 12.2)	5.1 (−16.8; 27.0)	7.0 (−7.8; 21.9)	13.9 (−6.8; 34.6)	23.5 (−8.4; 55.4)
Other vs. white/Caucasian	−6.4 (−66.8; 53.9)	−37.5 (−192.9; 117.9)	−23.1 (56.8; 10.5)	−8.3 (−45.1; 28.5)	−5.2 (−27.8; 17.4)	−2.2 (38.2; 33.7)	15.9 (−62.2; 93.9)
Peru							
Black vs. white/Caucasian	12.2 (−204.8; 229.2)	15.5 (−201.9; 225.9)	−4.4 (−22.7; 14.2)	79.9 (−54.3; 214.1)	−27.4 (−137.3; 82.4)	91.8 (−84.5; 267.9)	−3.7 (−15.6; 10.2)
Mixed vs. white/Caucasian	18.2 (−85.3; 49.0)	18.7 (−149.3; 186.5)	11.3 (−7.0; 29.7)	−9.1 (−36.6; 18.4)	−9.0 (−29.3; 11.2)	15.5 (−27.6; 58.7)	−104.0 (−204.6; −3.5)
Other vs. white/Caucasian	−96.1 (297.6; 105.4)	20.6 (−147.1; 188.8)	−14.1 (66.6; 38.2)	−56.2 (−123.3; 10.8)	13.4 (−41.7; 68.5)	83.6 (−2.4; 169.8)	−302.5 (−885.5; 280.3)
Venezuela							
Black vs. white/Caucasian	14.8 (−25.3; 54.9)	−51.4 (−139.2; 36.2)	−5.6 (−20.2; 8.9)	−6.3 (−29.5; 16.9)	−4.7 (−21.3; 11.9)	22.2 (−1.9; 46.3)	19.7 (61.5; 100.8)
Mixed vs. white/Caucasian	18.1 (1.7; 34.4)	−8.1 (−44.5; 28.3)	6.3 (−2.4; 15.0)	13.9 (2.9; 25.0)	10.8 (1.7; 19.8)	12.1 (1.7; 22.5)	11.2 (−16.7; 39.1)
Other vs. white/Caucasian	2.1 (−24.3; 28.6)	10.2 (−58.3; 78.7)	10.5 (−7.0; 28.0)	0.2 (−18.0; 17.7)	3.4 (−10.6; 17.4)	20.2 (1.1; 39.3)	22.8 (−33.1; 79.3)
Overall							
Black vs. white/Caucasian	41.0 (14.9; 67.1)	−1.8 (−56.3; 52.6)	12.9 (−0.4; 26.3)	−5.3 (−19.5; 8.8)	4.6 (−3.1; 12.4)	11.7 (−4.5; 27.9)	−6.4 (−38.0; 25.2)
Mixed vs. white/Caucasian	−8.4 (19.6; 2.7)	−15.2 (−37.0; 6.5)	−0.32 (−6.7; 6.1)	−10.1 (−17.1; −3.1)	1.3 (−2.7; 5.4)	−3.1 (−11.4; 5.2)	−11.2 (−24.3; 1.9)
Other vs. white/Caucasian	44.5 (19.9; 69.2)	8.0 (−38.5; 54.6)	47.7 (30.9; 64.4)	24.1 (10.0; 38.1)	20.9 (12.1; 29.7)	40.1 (23.8; 56.3)	7.1 (−17.5; 31.7)

Multilevel linear regression models, including region and cities as random effects, adjusted for sex, age, socioeconomic and education level. Reference: white/Caucasian; CI: confidence interval.

**Table 6 ijerph-17-05587-t006:** Adjusted analyses (*b* coefficient (95% CI)) between socioeconomic level and sedentary behavior by country for specific-domains.

Country	Computer at Home	Videogame Use	Reading	Socializing or Listen to Music	Talking on the Telephone	Watching TV	Riding in an Automobile
Argentina							
Middle vs. low	−8.4 (−26.3; 9.5)	6.6 (−22.4; 35.6)	0.2 (−11.0; 11.4)	−12.1 (−27.7; 3.5)	−1.8 (−10.1; 6.5)	−4.9 (−19.8; 9.8)	−9.1 (−33.9; 15.6)
High vs. low	−7.7 (−49.5; 34.1)	−92.1 (−156.3; −28.0)	−5.7 (−32.3; 20.8)	−4.6 (−46.8; 37.5)	−9.3 (−31.6; 12.9)	−28.4 (−63.8; 6.9)	7.4 (−68.9; 83.6)
Brazil							
Middle vs. low	−15.8 (−49.3; 17.8)	25.3 (−25.1; 75.7)	7.6 (−15.9; 31.1)	−4.8 (−21.5; 11.9)	−16.8 (−27.0; 6.6)	−36.2 (−58.4; −14.0)	−13.8 (−30.9; 3.2)
High vs. low	34.4 (−29.6; 98.4)	73.3 (−38.1; 184.6)	76.4 (26.6; 126.3)	28.6 (−5.2; 62.5)	−1.7 (−26.4; 22.9)	10.2 (−40.0; 60.5)	−2.4 (−26.8; 21.9)
Chile							
Middle vs. low	−9.1 (−32.9; 14.7)	−7.4 (−45.8; 30.9)	4.4 (−8.9; 17.8)	−0.3 (−15.7; 15.3)	−6.7 (−18.9; 5.4)	−6.2 (−21.5; 9.0)	6.9 (−14.8; 28.6)
High vs. low	31.0 (−20.2; 82.3)	−13.9 (−83.9; 55.9)	−6.3 (−28.9; 16.3)	0.47 (−33.6; 34.6)	−15.0 (−39.4; 9.2)	−12.1 (−49.1; 24.8)	5.6 (−36.2; 47.4)
Colombia							
Middle vs. low	−4.0 (−29.8; 21.7)	−10.2 (−43.8; 23.2)	7.1 (−5.5; 19.7)	−1.7 (−18.2; 14.7)	8.7 (−3.4; 20.9)	7.7 (−10.3; 25.7)	−20.2 (−56.5; 15.9)
High vs. low	−41.5 (−93.4; 10.4)	−25.5 (−93.4; 42.4)	−11.1 (−36.7; 14.6)	−19.8 (−54.3; 14.6)	5.3 (−16.8; 27.4)	−16.4 (−53.9; 21.1)	−2.6 (−69.8: 64.5)
Costa Rica							
Middle vs. low	−8.9 (−82.6; 64.8)	−58.9 (−153.1; 35.4)	10.9 (−29.5; 51.3)	−0.1 (−30.4; 30.5)	−0.4 (−14.4; 13.6)	−23.5 (−69.6; 22.6)	19.3 (−46.5; 85.1)
High vs. low	−38.4 (−138.5; 61.6)	−28.9 (−233.8; 175.9)	−8.7 (−59.1; 41.7)	18.4 (−37.5; 74.4)	16.1 (−17.7; 49.9)	−58.5 (−141.4; 24.3)	25.2 (−91.1; 141.6)
Ecuador							
Middle vs. low	5.7 (−10.4; 21.8)	−7.6 (−31.7; 16.3)	−2.8 (−12.3; 6.7)	2.5 (−8.3; 13.5)	0.9 (−6.9; 8.8)	2.6 (−7.9; 13.2)	2.2 (−20.1; 24.5)
High vs. low	12.6 (−10.6; 35.8)	−9.7 (−51.4; 31.8)	−8.9 (−25.9; 8.0)	13.8 (−4.7; 32.4)	−4.7 (−17.5; 8.1)	12.2 (−4.5; 29.0)	25.9 (−5.6; 57.6)
Peru							
Middle vs. low	−16.7 (−49.2; 15.8)	−21.2 (−58.3; 15.9)	−0.6 (−8.3; 7.1)	−1.9 (−14.3; 10.4)	−14.6 (−25.6; −3.7)	−7.6 (−27.8; 12.5)	11.7 (−67.2; 90.6)
High vs. low	37.9 (−5.4; 81.2)	20.5 (−24.9; 66.0)	14.8 (3.7; 25.8)	20.1 (−1.6; 39.5)	−6.7 (−20.7; 7.3)	−6.8 (−19.2; 32.9)	−13.4 (−86.6; 59.8)
Venezuela							
Middle vs. low	−2.1 (−18.9; 14.8)	1.9 (−32.2; 37.1)	9.0 (−0.9; 18.9)	−4.9 (−17.9; 8.2)	7.5 (−2.1; 17.3)	−2.4 (−14.5; 9.7)	10.1 (−18.4; 38.7)
High vs. low	8.5 (−19.1; 36.1)	64.0 (5.3; 122.7)	2.3 (−12.6; 17.1)	20.8 (−0.1; 41.6)	29.9 (13.8; 46.0)	9.9 (−11.2; 31.0)	12.7 (−30.9; 56.3)
Overall							
Middle vs. low	6.4 (−3.9; 16.7)	4.9 (−10.9; 20.7)	11.7 (5.7; 17.6)	4.2 (−1.7; 10.1)	−3.2 (−7.0; 0.5)	−0.4 (−8.0; 7.2)	−1.4 (−12.2; 9.3)
High vs. low	12.4 (−4.5; 29.3)	18.6 (−9.11; 46.4)	15.1 (5.3; 24.9)	12.9 (2.2; 23.5)	3.1 (−4.2; 10.6)	3.1 (−11.1; 17.2)	7.5 (−10.2; 25.1)

Multilevel linear regression models, including region and cities as random effects, adjusted for sex, age, ethnicity, and education level. Reference: low; CI: confidence interval.

**Table 7 ijerph-17-05587-t007:** Adjusted analyses (*b* coefficient (95% CI)) between education level and sedentary behavior by country for specific-domains.

Country	Computer at Home	Videogame Use	Reading	Socializing or Listen to Music	Talking on the Telephone	Watching TV	Riding in an Automobile
Argentina							
Middle vs. low	17.2 (−1.9; 36.4)	58.6 (18.3; 98.8)	40.9 (29.2; 52.7)	13.4 (−5.3; 32.2)	10.8 (0.9; 20.9)	−7.4 (−25.5; 10.6)	−8.5 (36.5; 19.4)
High vs. low	29.4 (−5.8; 64.6)	22.8 (−36.3; 81.9)	36.5 (18.6; 54.4)	20.4 (−19.6; 60.6)	3.4 (−15.4; 22.2)	−11.9 (−47.4; 23.5)	−8.5 (−40.8; 57.9)
Brazil							
Middle vs. low	−21.0 (−51.9; 9.9)	−47.3 (103.0; 8.3)	4.7 (−18.6; 27.9)	−18.1 (−33.8; −2.3)	3.0 (−7.0; 12.9)	−4.6 (−17.6; 26.9)	12.2 (−0.3; 24.8)
High vs. low	69.0 (1.8; 136.3)	−57.5 (−254.0; 138.9)	39.1 (−14.5; 92.7)	26.6 (−11.3; 64.6)	17.9 (−5.4; 41.4)	36.7 (−7.7; 81.0)	39.6 (10.3; 68.9)
Chile							
Middle vs. low	−1.0 (−26.2; 24.2)	−9.2 (−53.8; 35.3)	23.4 (9.6; 37.3)	−2.5 (−18.9; 13.9)	0.6 (−11.2; 12.4)	7.2 (−10.2; 24.6)	20.4 (−0.9; 41.8)
High vs. low	5.5 (−30.0; 41.0)	−0.3 (−57.5; 56.9)	8.0 (−7.3; 23.4)	19.4 (−7.4; 43.6)	18.9 (−1.7; 39.6)	−16.4 (−41.3; 8.4)	12.7 (−13.5; 39.0)
Colombia							
Middle vs. low	10.6 (−15.6; 36.7)	10.9 (−21,8; 43.7)	16.7 (3.0; 30.3)	9.4 (−8.1; 27.0)	−8.9 (−21.5; 3.7)	−4.4 (−24.5; 15.7)	24.1 (−15.1; 63.4)
High vs. low	19.0 (−16.3; 54.4)	−6.6 (−64.2; 51.1)	15,3 (−1.3; 31.8)	6.3 (−18.2; 30.8)	9.7 (−10.5; 29.9)	−12.8 (−38.1; 12.4)	27.2 (−23.0; 77.3)
Costa Rica							
Middle vs. low	10.5 (−62.1; 83.1)	15.2 (−106.9; 137.2)	22.5 (−23.6; 68.7)	28.5 (−14.4; 71.6)	21.2 (−1.3; 43.7)	−15.9 (−79.3; 47.4)	−25.5 (−115.4; 64.4)
High vs. low	−25.6 (−116.6; 65.4)	−26.7 (−189.0; 135.5)	20.8 (−46.8; 88.5)	−49.0 (−109.3; 11.3)	43.9 (15.1; 72.8)	−46.7 (−138.0; 44.6)	−50.2 (−151.1; 51.0)
Ecuador							
Middle vs. low	21.3 (1.6; 41.1)	44.4 (14.3; 74.5)	27.6 (13.2; 42.1)	7.0 (−9.0; 23.0)	−0.9 (−12.6; 10.8)	4.9 (−11.1; 20.9)	48.8 (16.9; 80.7)
High vs. low	33.2 (9.0; 57.4)	−7.8 (−35.9; 51.4)	8.0 (−7.8; 23.9)	7.8 (−14,5; 30.2)	13.9 (−1.4; 29.3)	1.6 (−20.3; 23.6)	15.6 (−16.4; 47.5)
Peru							
Middle vs. low	−5.8 (−50.2; 38.5)	−13.1 (−54.7; 28.5)	−1.1 (−9.6; 7.3)	−7.1 (−21.0; 6.8)	8.3 (−3.9; 20.7)	−15.1 (−36.5; 6.2)	27.7 (−52.8; 108.2)
High vs. low	−4.3 (−76.2; 67.6)	−0.1 (−159.7; 159.9)	5.1 (−13.6; 23.9)	−36.2 (−65.2; −7.1)	−4.1 (−26.4; 18.1)	−32.4 (−79.2; −14.4)	−74.6 (−218.4; 69.3)
Venezuela							
Middle vs. low	−16.5 (−34.7; 1.6)	0.6 (−36.9; 38.1)	−5.8 (−14.0; 2.3)	−12.3 (−27.2; 2.6)	−4.2 (−15.3; 6.8)	−0.4 (−15.1; 14.3)	−16.5 (−51.9; 18.8)
High vs. low	15.4 (−1.5; 32.2)	35.2 (0.4; 70.9)	13.9 (3.7; 24.1)	8.6 (−4.1; 21.4)	6.3 (−3.9; 16.5)	−0.8 (−13.4; 11.9)	10.9 (−18.2; 40.0)
Overall							
Middle vs. low	7.0 (−3.5; 17.5)	−2.3 (−19.9; 15.2)	9.8 (3.6; 15.9)	−5.8 (−12.1; 0.4)	6.8 (2.8; 10.8)	−0.8 (−9.2; 7.5)	−1.2 (−12.4; 10.1)
High vs. low	15.5 (0.1; 30.9)	−17.4 (−48.4; 13.6)	16.6 (6.5; 26.8)	−3.1 (−13.7; 7.5)	10.3 (3.5; 16.9)	−14.8 (−27.7; −1.8)	4.7 (−11.2; 20.7)

Multilevel linear regression models, including region and cities as random effects, adjusted for sex, age, ethnicity, socioeconomic and education level. Reference: low; CI: confidence interval.

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
