# Peer review of "Socio-Demographic Correlates of Total and Domain-Specific Sedentary Behavior in Latin America: A Population-Based Study"

_ijerph, 2020, doi:10.3390/ijerph17155587_

Round 1
Reviewer 1 Report
This is an interesting cross-sectional study about the correlates of sedentary behavior in eight Latin American countries.
Some minor suggestions are:
1) Abstract: please include the study design in the methods section of the abstract
2) Abstract vs. Methods: in the abstract (line 36) you state that data was collected from September 2014 to February 2015. However, in the methods section (line 95) you state that "Data collection occurred between September 2014 and October 2015." Thus, the time frames are disaccording. Please modify.
3) Introduction (line 61/62): I would suggest to insert Ref. 13 (Tremblay et al, 2014) instead of Ref. 4 for the definition of SB
4) Methods: It would be of interest why the Salmon Sedentary Behaviour Questionnaire was chosen as the adequate instrument for assessing self-reported SB in this population. In addition, psychometric testing data exists on the reliability and validity of the questionnaire compared to objective SB assessment with ActiGraph.
5) Discussion: You could strengthen your discussion by including the aspect of objective vs. subjective assessment of SB.
6) Discussion: Although the questionnaire applied in this study does not assess SB at work, this is an interesting aspect and a renowned domain of SB. You should discuss this aspect more in detail (than in line 263-265).
Author Response
International Journal of Environmental Research and Public Health
June 15th, 2020
Prof. Dr. Paul B. Tchounwou
Editor-in-Chief - International Journal of Environmental Research and Public Health
Thank you for considering our manuscript entitled " Socio-demographic correlates of total and domain-specific sedentary behavior in Latin America: A population-based study”, which has been sent for review at the International Journal of Environmental Research and Public Health.
We have read the reviewer’s comments and addressed each of them carefully. These changes are highlighted in red in the response to the reviewers but also in the manuscript. A copy with track changes and a clean copy has been submitted to the journal.
Overall, we feel that revisions made have resulted in a stronger manuscript and that the findings represent a significant contribution to the literature on sedentary behaviour surveillance and associations between sociodemographic factors and both total and domain-specific SB. We trust that you will feel the same and we look forward to your editorial decision.
Yours sincerely,
Dr. Gerson Luis de Moraes Ferrari
Corresponding author (on behalf of the authors)
We would like to thank the reviewers for taking the time to review our manuscript and we appreciate their constructive and thoughtful comments. Based on these suggestions, we have revised our manuscript accordingly. Please find below our point-by-point responses to the comments and suggestions.
Reviewer: 1
General comments
This is an interesting cross-sectional study about the correlates of sedentary behavior in eight Latin American countries.
Author’s response: Thank you for your feedback and positive comment. Please find below our point-by-point responses to your comments and suggestions.
Abstract
- Abstract: please include the study design in the methods section of the abstract.
Author’s response: In line with your suggestion, we have now included this information in the abstract.
- Abstract vs. Methods: in the abstract (line 36) you state that data was collected from September 2014 to February 2015. However, in the methods section (line 95) you state that "Data collection occurred between September 2014 and October 2015." Thus, the time frames are disaccording. Please modify.
Author’s response: Thank you for pointing this discrepancy out. We have made the necessary correction in the methods section.
Introduction (line 61/62)
- I would suggest to insert Ref. 13 (Tremblay et al, 2014) instead of Ref. 4 for the definition of SB.
Author’s response: As you have suggested, we have revised the reference used for the definition of sedentary behaviour.
Methods
- It would be of interest why the Salmon Sedentary Behaviour Questionnaire was chosen as the adequate instrument for assessing self-reported SB in this population. In addition, psychometric testing data exists on the reliability and validity of the questionnaire compared to objective SB assessment with ActiGraph.
Author’s response: To address your comments pertaining to why this questionnaire was selected and relevant psychometric properties, we have added this information in the methods section.
Discussion
- You could strengthen your discussion by including the aspect of objective vs. subjective assessment of SB.
Author’s response: Thanks for the suggestion. We have added a section in the discussion highlighting issues surrounding objective vs. subjective measurement of SB.
- Although the questionnaire applied in this study does not assess SB at work, this is an interesting aspect and a renowned domain of SB. You should discuss this aspect more in detail (than in line 263-265).
Author’s response: In line with your suggestion, we have added this information in the discussion.

Reviewer 2 Report
Thank you for your contribution to this important topic!
The authors promise a twofold analysis: Socio-demographic correlates of total and domain-specific sedentary behavior. However, in the abstract they summarize correlates of domain-specific behavior only. I would encourage adding a summary of correlates of total sedentary behavior as well because total sedentary time is what relates to health outcomes.
The authors´ claim in line 66 are mistaken. Watching TV has not been shown in recent studies to be a major aspect of SB. References 6 and 7 do not show this. Some of the studies summarized in ref 6 no not even distinguish among indoor SBs, others do have a category "screen time" but in 2019 this does not equate to TV watching. Ref 7 also measures screen time only.
A few words about the reporting of statistics, both regarding review of the prior literature in the introduction and reporting of present findings. I would encourage a clear explanation whether the reported regression coefficients are standardized or not (b´s or β´s). If reporting b´s , it should be made clear what the predicted reference category is (e.g. 1 hour increase in SB). When reporting β´s, there should be a discussion of effect size, e.g. what cutoffs the authors used for small, medium, or large effect sizes. In the present study the authors claim to be presenting β-coefficients. The international standard is that β refers to a standardized coefficient. However the reported values make it more likely that these are in fact unstandardized coefficients which should be denoted b (a small Latin "b" in italics).
With a large N as is typical of this type of study, null hypothesis testing can be very misleading as even miniscule effects can be found significant. For this reason β´s should be reported, again with discussion of the effect size.
Finally, in the discussion, there should also be some benchmarking of regression coefficients with the magnitude of effect that has been shown in the literature to affect health outcomes, often a reduction in SB of about 40 - 60 minutes per day is used. With a large N as is typical of this type of study, null hypothesis testing can be very misleading as even miniscule effects can be found significant.
The authors should justify on clinical grounds their selection of 275 min/day as the cutoff between high and low SB , i.e. in relation to known health risks related to SB. Alternatively, a different cutoff should be selected that is justifiable on clinical grounds.
Most studies in this area present hierarchical regression analyses to control for the effects of demographic variables including age, gender, and SES. I would encourage such an analysis here especially for the country-by-country comparison. Without it, readers including public health officials are left to guess whether differences are merely due to variations in gender and age of participants included in the study or due to the prevailing SES in their country.
In this context is it confusing why the authors would present a multilevel analysis adjusting for ethnicity. Ethnicity is a common factor causing health disparities and its adjustment can serve to hide this concerning finding. Instead, the authors should be encouraged to present an analysis whether ethnicity is a risk factor so that public resources can be directed to those who need them most. Based on the results in Table 1, it would appear that Black and "other" ethnicity is indeed a risk factor here. Again, this should be further analyzed and described.
From line 150 until 206, the authors are duplicating a lot of table information in the text. The redundant presentation of a lot of number in text when those numbers appear in tables is generally discouraged. Instead, authors should be encouraged to discuss important findings from the tables in general terms with particular emphasis on effect sizes and patterns, such as ethnic disparities, that casual inspection of tables may not reveal.
Tables 2 through 5 need clarification as to whether standardized or non-standardized regression coefficients are reported. If these are non-standardized coefficients, they should minimally be starred to indicate their statistical significance and, more importantly, their effect size.
In line 225, the authors again claim that TV watching is of particular concern which is not accurate (see above).
The discussion also suffers from the same problem as the abstract in that there is little emphasis on correlates of total SB and much of differences in particular types of SB. While there is some evidence of marginal differences in the health effects based on type of SB, there is extensive evidence that excessive SB is harmful. I would therefore encourage detailed analysis of correlates of total SB time including references to clinical benchmarking, i.e. what magnitude of difference in total SB time is expected to make a real difference regarding health outcomes.
Finally, there is no discussion of the ethnic disparities evident in Table 1. Again, these should be elucidated through further analysis, such as hierarchical regression controlling for SES, and then discussed.
With these changes, the manuscript can make an important contribution to the literature on SB.
Author Response
International Journal of Environmental Research and Public Health
June 15th, 2020
Prof. Dr. Paul B. Tchounwou
Editor-in-Chief - International Journal of Environmental Research and Public Health
Thank you for considering our manuscript entitled " Socio-demographic correlates of total and domain-specific sedentary behavior in Latin America: A population-based study”, which has been sent for review at the International Journal of Environmental Research and Public Health.
We have read the reviewer’s comments and addressed each of them carefully. These changes are highlighted in red in the response to the reviewers but also in the manuscript. A copy with track changes and a clean copy has been submitted to the journal.
Overall, we feel that revisions made have resulted in a stronger manuscript and that the findings represent a significant contribution to the literature on sedentary behaviour surveillance and associations between sociodemographic factors and both total and domain-specific SB. We trust that you will feel the same and we look forward to your editorial decision.
Yours sincerely,
Dr. Gerson Luis de Moraes Ferrari
Corresponding author (on behalf of the authors)
We would like to thank the reviewers for taking the time to review our manuscript and we appreciate their constructive and thoughtful comments. Based on these suggestions, we have revised our manuscript accordingly. Please find below our point-by-point responses to the comments and suggestions.
Reviewer: 2
General comments
Thank you for your contribution to this important topic!
Author’s response: Thank you for your feedback and comments. Please find below our point-by-point responses to your comments and suggestions.
- The authors promise a twofold analysis: Socio-demographic correlates of total and domain-specific sedentary behavior. However, in the abstract they summarize correlates of domain-specific behavior only. I would encourage adding a summary of correlates of total sedentary behavior as well because total sedentary time is what relates to health outcomes.
Author’s response: Thanks for the suggestion – we have now included findings pertaining to total SB in the abstract.
- The authors´ claim in line 66 are mistaken. Watching TV has not been shown in recent studies to be a major aspect of SB. References 6 and 7 do not show this. Some of the studies summarized in ref 6 no not even distinguish among indoor SBs, others do have a category "screen time" but in 2019 this does not equate to TV watching. Ref 7 also measures screen time only.
Author’s response: Thank you for pointing this out. We have corrected it.
3-4. A few words about the reporting of statistics, both regarding review of the prior literature in the introduction and reporting of present findings. I would encourage a clear explanation whether the reported regression coefficients are standardized or not (b´s or β´s). If reporting b´s, it should be made clear what the predicted reference category is (e.g. 1 hour increase in SB). When reporting β´s, there should be a discussion of effect size, e.g. what cutoffs the authors used for small, medium, or large effect sizes. In the present study the authors claim to be presenting β-coefficients. The international standard is that β refers to a standardized coefficient. However the reported values make it more likely that these are in fact unstandardized coefficients which should be denoted b (a small Latin "b" in italics).
With a large N as is typical of this type of study, null hypothesis testing can be very misleading as even miniscule effects can be found significant. For this reason β´s should be reported, again with discussion of the effect size.
Author’s response: We report unstandardized coefficients – this has now been stipulated in our data analyses section. We have adjusted the nomenclature throughout the text according to your recommendation. We believe that the interpretation of unstandardized coefficients is more clear and applicable to understanding effects with regard to differences in total and domain-specific sedentary time based on sociodemographic factors. We also adjusted the description of the results to improve the comprehension. We also agree that with a larger N, the p-values can be affected. For this, we focused on the unstandardized coefficient values and not on the “statistical significance” per se.
- Finally, in the discussion, there should also be some benchmarking of regression coefficients with the magnitude of effect that has been shown in the literature to affect health outcomes, often a reduction in SB of about 40 - 60 minutes per day is used. With a large N as is typical of this type of study, null hypothesis testing can be very misleading as even miniscule effects can be found significant.
Author’s response: In line with your comment, we have now included some discussion as to the dose-response relationships that exist between SB and health outcomes.
- The authors should justify on clinical grounds their selection of 275 min/day as the cutoff between high and low SB, i.e. in relation to known health risks related to SB. Alternatively, a different cutoff should be selected that is justifiable on clinical grounds.
Author’s response: Thank you for pointing this out. We used the median cutoff (275 min/day) point because there are currently no specific guidelines outlining the recommended time spent in total or domain-specific sedentary behavior for adults (references 1 to 3 - bellow). It should also be noted that there is not consensus on the cut-off that should be used to categorize individuals as “sitting too much”. Previous manuscripts that analyzed the specific domains of sedentary behavior used the median as the cutoff point to distinguish between high and low levels of SB (references 3 to 6 – bellow). We have provided additional information in the statistical analysis section justifying the use of the median as a cutoff point.
1 - https://www.annualreviews.org/doi/10.1146/annurev-publhealth-040119-094201
2 - https://pubmed.ncbi.nlm.nih.gov/28599680/
3 - https://pubmed.ncbi.nlm.nih.gov/29891615/
4 - https://www.ncbi.nlm.nih.gov/pmc/articles/PMC3950247/
6 - https://content.apa.org/record/2003-01659-007
- Most studies in this area present hierarchical regression analyses to control for the effects of demographic variables including age, gender, and SES. I would encourage such an analysis here especially for the country-by-country comparison. Without it, readers including public health officials are left to guess whether differences are merely due to variations in gender and age of participants included in the study or due to the prevailing SES in their country.
Author’s response: Thank you for your suggestion. We agree and in our revised manuscript, we have presented findings from the regression models adjusted for the independent variables (sex, age, ethnicity, socioeconomic and education level).
- In this context is it confusing why the authors would present a multilevel analysis adjusting for ethnicity. Ethnicity is a common factor causing health disparities and its adjustment can serve to hide this concerning finding. Instead, the authors should be encouraged to present an analysis whether ethnicity is a risk factor so that public resources can be directed to those who need them most. Based on the results in Table 1, it would appear that Black and "other" ethnicity is indeed a risk factor here. Again, this should be further analyzed and described.
Author’s response: Thank you for the suggestion. We have added a new table (table 4) to present the association between ethnicity and sedentary behavior adjusted for sex, age, socioeconomic and education level.
- From line 150 until 206, the authors are duplicating a lot of table information in the text. The redundant presentation of a lot of number in text when those numbers appear in tables is generally discouraged. Instead, authors should be encouraged to discuss important findings from the tables in general terms with particular emphasis on effect sizes and patterns, such as ethnic disparities, that casual inspection of tables may not reveal.
Author’s response: We thank the reviewer for this comment. We have now focused our results’ description on the size of the difference in clinically relevant values.
- Tables 2 through 5 need clarification as to whether standardized or non-standardized regression coefficients are reported. If these are non-standardized coefficients, they should minimally be starred to indicate their statistical significance and, more importantly, their effect size.
Author’s response: We thank the reviewer for this comment. We reported unstandardized coefficients and as previously noted, we have adjusted the nomenclature in the statistics section as well as in the tables (as suggested). Also, we have now focused our results’ description on the size of the difference in clinically relevant values.
- In line 225, the authors again claim that TV watching is of particular concern which is not accurate (see above).
Author’s response: To address your concern, we have reworded and revised this statement accordingly.
- The discussion also suffers from the same problem as the abstract in that there is little emphasis on correlates of total SB and much of differences in particular types of SB. While there is some evidence of marginal differences in the health effects based on type of SB, there is extensive evidence that excessive SB is harmful. I would therefore encourage detailed analysis of correlates of total SB time including references to clinical benchmarking, i.e. what magnitude of difference in total SB time is expected to make a real difference regarding health outcomes.
Author’s response: As per your suggestion, we have added a new analysis (association of total sedentary time with each independent variable separated by country) presented in Table 7. In addition, we have inserted a paragraph to address and discuss the findings with respect to total sedentary behavior.
- Finally, there is no discussion of the ethnic disparities evident in Table 1. Again, these should be elucidated through further analysis, such as hierarchical regression controlling for SES, and then discussed. With these changes, the manuscript can make an important contribution to the literature on SB.
Author’s response: Thank you for your suggestion. We have added a new table that shows regression findings concerning the association between ethinicity and domain-specific sedentary behavior by country.

Round 2
Reviewer 2 Report
Thank you for giving me another look at this manuscript!
Here is my feedback to the changes the authors made: For ease of communication, I will reprint the authors’ text with my new comments from 6-3-2020 inserted in italics:
General comments
Thank you for your contribution to this important topic!
Author’s response: Thank you for your feedback and comments. Please find below our point-by-point responses to your comments and suggestions.
- The authors promise a twofold analysis: Socio-demographic correlates of total and domain-specific sedentary behavior. However, in the abstract they summarize correlates of domain-specific behavior only. I would encourage adding a summary of correlates of total sedentary behavior as well because total sedentary time is what relates to health outcomes.
Author’s response: Thanks for the suggestion – we have now included findings pertaining to total SB in the abstract.
6/30/2020 comment: successfully addresses the concern.
- The authors´ claim in line 66 are mistaken. Watching TV has not been shown in recent studies to be a major aspect of SB. References 6 and 7 do not show this. Some of the studies summarized in ref 6 no not even distinguish among indoor SBs, others do have a category "screen time" but in 2019 this does not equate to TV watching. Ref 7 also measures screen time only.
6/30/2020 comment: The only specific example of SB given in this section remains ‘TV watching’. (line 67). Given that smartphone and related social media use is prevalent in all countries surveyed, the authors will want to expand that example.
Author’s response: Thank you for pointing this out. We have corrected it.
3-4. A few words about the reporting of statistics, both regarding review of the prior literature in the introduction and reporting of present findings. I would encourage a clear explanation whether the reported regression coefficients are standardized or not (b´s or β´s). If reporting b´s, it should be made clear what the predicted reference category is (e.g. 1 hour increase in SB). When reporting β´s, there should be a discussion of effect size, e.g. what cutoffs the authors used for small, medium, or large effect sizes. In the present study the authors claim to be presenting β-coefficients. The international standard is that β refers to a standardized coefficient. However the reported values make it more likely that these are in fact unstandardized coefficients which should be denoted b (a small Latin "b" in italics).
With a large N as is typical of this type of study, null hypothesis testing can be very misleading as even miniscule effects can be found significant. For this reason β´s should be reported, again with discussion of the effect size.
Author’s response: We report unstandardized coefficients – this has now been stipulated in our data analyses section. We have adjusted the nomenclature throughout the text according to your recommendation. We believe that the interpretation of unstandardized coefficients is more clear and applicable to understanding effects with regard to differences in total and domain-specific sedentary time based on sociodemographic factors. We also adjusted the description of the results to improve the comprehension. We also agree that with a larger N, the p-values can be affected. For this, we focused on the unstandardized coefficient values and not on the “statistical significance” per se.
6/30/2020 comment: While β has been eliminated from the table, it continues to appear on line 43,and possibly elsewhere.
- Finally, in the discussion, there should also be some benchmarking of regression coefficients with the magnitude of effect that has been shown in the literature to affect health outcomes, often a reduction in SB of about 40 - 60 minutes per day is used. With a large N as is typical of this type of study, null hypothesis testing can be very misleading as even miniscule effects can be found significant.
Author’s response: In line with your comment, we have now included some discussion as to the dose-response relationships that exist between SB and health outcomes.
6/30/2020 comment: successfully addresses the concern.
- The authors should justify on clinical grounds their selection of 275 min/day as the cutoff between high and low SB, i.e. in relation to known health risks related to SB. Alternatively, a different cutoff should be selected that is justifiable on clinical grounds.
Author’s response: Thank you for pointing this out. We used the median cutoff (275 min/day) point because there are currently no specific guidelines outlining the recommended time spent in total or domain-specific sedentary behavior for adults (references 1 to 3 - bellow). It should also be noted that there is not consensus on the cut-off that should be used to categorize individuals as “sitting too much”. Previous manuscripts that analyzed the specific domains of sedentary behavior used the median as the cutoff point to distinguish between high and low levels of SB (references 3 to 6 – bellow). We have provided additional information in the statistical analysis section justifying the use of the median as a cutoff point.
1 - https://www.annualreviews.org/doi/10.1146/annurev-publhealth-040119-094201
2 - https://pubmed.ncbi.nlm.nih.gov/28599680/
3 - https://pubmed.ncbi.nlm.nih.gov/29891615/
4 - https://www.ncbi.nlm.nih.gov/pmc/articles/PMC3950247/
6 - https://content.apa.org/record/2003-01659-007
6/30/2020 comment: successfully addresses the concern.
- Most studies in this area present hierarchical regression analyses to control for the effects of demographic variables including age, gender, and SES. I would encourage such an analysis here especially for the country-by-country comparison. Without it, readers including public health officials are left to guess whether differences are merely due to variations in gender and age of participants included in the study or due to the prevailing SES in their country.
Author’s response: Thank you for your suggestion. We agree and in our revised manuscript, we have presented findings from the regression models adjusted for the independent variables (sex, age, ethnicity, socioeconomic and education level).
6/30/2020 comment: concern was successfully addressed except regarding ethnicity (see below).
- In this context is it confusing why the authors would present a multilevel analysis adjusting for ethnicity. Ethnicity is a common factor causing health disparities and its adjustment can serve to hide this concerning finding. Instead, the authors should be encouraged to present an analysis whether ethnicity is a risk factor so that public resources can be directed to those who need them most. Based on the results in Table 1, it would appear that Black and "other" ethnicity is indeed a risk factor here. Again, this should be further analyzed and described.
Author’s response: Thank you for the suggestion. We have added a new table (table 4) to present the association between ethnicity and sedentary behavior adjusted for sex, age, socioeconomic and education level.
6/30/2020 comment: When analyzing for ethnicity as a predictor it is inappropriate to adjust for socioeconomic and education level variables because in many societies these are correlated with ethnicity. Regressing them out pretends all ethnicities have equal income and educational opportunities which is factually incorrect in most countries and serves to hide the impact of ethnicity on health outcomes. Therefore, ethnicity as a predictor should retain variability due to socioeconomic and educational factors.
- From line 150 until 206, the authors are duplicating a lot of table information in the text. The redundant presentation of a lot of number in text when those numbers appear in tables is generally discouraged. Instead, authors should be encouraged to discuss important findings from the tables in general terms with particular emphasis on effect sizes and patterns, such as ethnic disparities, that casual inspection of tables may not reveal.
Author’s response: We thank the reviewer for this comment. We have now focused our results’ description on the size of the difference in clinically relevant values.
6/30/2020 comment: I would suggest making Table 7 the new Table 2 since Table 7 pertains to overall SB and old Tables 2 through 6 to domain specific SB. This would also lay the conceptual foundation for covariate adjustments in subsequent analyses.
Also, on line 128 the authors now report the ICC for ‘talking on the telephone’ is .06 which means this is not a reliable category (scores are essentially random). Therefore, this SB domain should be eliminated from all analyses.
The early part of the results section, until line 193, still contains a lot of duplicated table numbers and I would encourage the authors to edit that part of the results section to look more like lines 194ff.
- Tables 2 through 5 need clarification as to whether standardized or non-standardized regression coefficients are reported. If these are non-standardized coefficients, they should minimally be starred to indicate their statistical significance and, more importantly, their effect size.
Author’s response: We thank the reviewer for this comment. We reported unstandardized coefficients and as previously noted, we have adjusted the nomenclature in the statistics section as well as in the tables (as suggested). Also, we have now focused our results’ description on the size of the difference in clinically relevant values.
6/30/2020 comment: Concern successfully addressed.
- In line 225, the authors again claim that TV watching is of particular concern which is not accurate (see above).
Author’s response: To address your concern, we have reworded and revised this statement accordingly.
6/30/2020 comment: Concern somewhat addressed, I still would encourage the authors to acknowledge the role of smartphone and social media use in causing SB especially among younger age cohorts..
- The discussion also suffers from the same problem as the abstract in that there is little emphasis on correlates of total SB and much of differences in particular types of SB. While there is some evidence of marginal differences in the health effects based on type of SB, there is extensive evidence that excessive SB is harmful. I would therefore encourage detailed analysis of correlates of total SB time including references to clinical benchmarking, i.e. what magnitude of difference in total SB time is expected to make a real difference regarding health outcomes.
Author’s response: As per your suggestion, we have added a new analysis (association of total sedentary time with each independent variable separated by country) presented in Table 7. In addition, we have inserted a paragraph to address and discuss the findings with respect to total sedentary behavior.
6/30/2020 comment: The discussion section still lacks a discussion that focusses clearly on correlates of the amount of SB. Again, there is little evidence SB domains impact health; there is lots of evidence overall SB amount impacts health. I would suggest clearly and cleanly discussing correlates of overall SB time first. Once that discussion is done, discussion of SB domains can be done but this should be de-emphasized and qualified since, again, there is comparatively little evidence that a person’s health is impacted by what they do while sedentary.
- Finally, there is no discussion of the ethnic disparities evident in Table 1. Again, these should be elucidated through further analysis, such as hierarchical regression controlling for SES, and then discussed. With these changes, the manuscript can make an important contribution to the literature on SB.
Author’s response: Thank you for your suggestion. We have added a new table that shows regression findings concerning the association between ethinicity and domain-specific sedentary behavior by country.
6/30/2020 comment: Clear (or really any) discussion of ethnicity as a correlate of total SB time is lacking. The same is true for all other correlates. E.g. What factors put ethnic minorities at risk for increased SB time, e.g. residence in neighborhoods where walking or exercise is unsafe, lack of access to gyms, etc. Same is true for all other factors: Why are men at greater risk etc.
Author Response
International Journal of Environmental Research and Public Health
July 4th, 2020
Prof. Dr. Paul B. Tchounwou
Editor-in-Chief - International Journal of Environmental Research and Public Health
Thank you for considering our manuscript entitled "Socio-demographic correlates of total and domain-specific sedentary behavior in Latin America: A population-based study”, which has been sent for review at the International Journal of Environmental Research and Public Health.
We have read the reviewer’s comments and addressed each of them carefully. These changes are highlighted in red in the manuscript. A copy with tracked changes has been submitted to the journal.
Overall, we feel that revisions made have resulted in a stronger manuscript and that the findings represent a significant contribution to the literature on sedentary behaviour surveillance and associations between sociodemographic factors and both total and domain-specific SB. We trust that you will feel the same and we look forward to your editorial decision.
Yours sincerely,
Dr. Gerson Luis de Moraes Ferrari
Corresponding author (on behalf of the authors)
We would like to thank the reviewers for taking the time to review our manuscript and we appreciate their constructive and thoughtful comments. Based on these suggestions, we have revised our manuscript accordingly. Please find below our point-by-point responses to the comments and suggestions.
Reviewer: 2
Thank you for giving me another look at this manuscript!
Here is my feedback to the changes the authors made: For ease of communication, I will reprint the authors’ text with my new comments from 6-3-2020 inserted in italics:
Author’s response: We have read the reviewer’s comments and addressed each of them carefully. These changes are highlighted in red in the manuscript
General comments
Thank you for your contribution to this important topic!
Author’s response: Thank you for your feedback and comments. Please find below our point-by-point responses to your comments and suggestions.
- The authors promise a twofold analysis: Socio-demographic correlates of total and domain-specific sedentary behavior. However, in the abstract they summarize correlates of domain-specific behavior only. I would encourage adding a summary of correlates of total sedentary behavior as well because total sedentary time is what relates to health outcomes.
Author’s response: Thanks for the suggestion – we have now included findings pertaining to total SB in the abstract.
6/30/2020 comment: successfully addresses the concern.
Author’s response: Thank you.
- The authors´ claim in line 66 are mistaken. Watching TV has not been shown in recent studies to be a major aspect of SB. References 6 and 7 do not show this. Some of the studies summarized in ref 6 no not even distinguish among indoor SBs, others do have a category "screen time" but in 2019 this does not equate to TV watching. Ref 7 also measures screen time only.
Author’s response: Thank you for pointing this out. We have corrected it.
6/30/2020 comment: The only specific example of SB given in this section remains ¨TV watching¨. (line 67). Given that smartphone and related social media use is prevalent in all countries surveyed, the authors will want to expand that example.
Author’s response: As you have suggested, we have now included additional examples in the Introduction section.
3-4. A few words about the reporting of statistics, both regarding review of the prior literature in the introduction and reporting of present findings. I would encourage a clear explanation whether the reported regression coefficients are standardized or not (b´s or β´s). If reporting b´s, it should be made clear what the predicted reference category is (e.g. 1 hour increase in SB). When reporting β´s, there should be a discussion of effect size, e.g. what cutoffs the authors used for small, medium, or large effect sizes. In the present study the authors claim to be presenting β-coefficients. The international standard is that β refers to a standardized coefficient. However the reported values make it more likely that these are in fact unstandardized coefficients which should be denoted b (a small Latin "b" in italics).
With a large N as is typical of this type of study, null hypothesis testing can be very misleading as even miniscule effects can be found significant. For this reason β´s should be reported, again with discussion of the effect size.
Author’s response: We report unstandardized coefficients – this has now been stipulated in our data analyses section. We have adjusted the nomenclature throughout the text according to your recommendation. We believe that the interpretation of unstandardized coefficients is more clear and applicable to understanding effects with regard to differences in total and domain-specific sedentary time based on sociodemographic factors. We also adjusted the description of the results to improve the comprehension. We also agree that with a larger N, the p-values can be affected. For this, we focused on the unstandardized coefficient values and not on the “statistical significance” per se.
6/30/2020 comment: While β has been eliminated from the table, it continues to appear on line 43, and possibly elsewhere.
Author’s response: Thank you for pointing this error out – we apologize for this. We have re-checked the manuscript and removed any occurrence of (β) throughout the manuscript.
- Finally, in the discussion, there should also be some benchmarking of regression coefficients with the magnitude of effect that has been shown in the literature to affect health outcomes, often a reduction in SB of about 40 - 60 minutes per day is used. With a large N as is typical of this type of study, null hypothesis testing can be very misleading as even miniscule effects can be found significant.
Author’s response: In line with your comment, we have now included some discussion as to the dose-response relationships that exist between SB and health outcomes.
6/30/2020 comment: successfully addresses the concern.
Author’s response: Thank you.
- The authors should justify on clinical grounds their selection of 275 min/day as the cutoff between high and low SB, i.e. in relation to known health risks related to SB. Alternatively, a different cutoff should be selected that is justifiable on clinical grounds.
Author’s response: Thank you for pointing this out. We used the median cutoff (275 min/day) point because there are currently no specific guidelines outlining the recommended time spent in total or domain-specific sedentary behavior for adults (references 1 to 3 - bellow). It should also be noted that there is not consensus on the cut-off that should be used to categorize individuals as “sitting too much”. Previous manuscripts that analyzed the specific domains of sedentary behavior used the median as the cutoff point to distinguish between high and low levels of SB (references 3 to 6 – bellow). We have provided additional information in the statistical analysis section justifying the use of the median as a cutoff point.
1 - https://www.annualreviews.org/doi/10.1146/annurev-publhealth-040119-094201
2 - https://pubmed.ncbi.nlm.nih.gov/28599680/
3 - https://pubmed.ncbi.nlm.nih.gov/29891615/
4 - https://www.ncbi.nlm.nih.gov/pmc/articles/PMC3950247/
6 - https://content.apa.org/record/2003-01659-007
6/30/2020 comment: successfully addresses the concern.
Author’s response: Thank you.
- Most studies in this area present hierarchical regression analyses to control for the effects of demographic variables including age, gender, and SES. I would encourage such an analysis here especially for the country-by-country comparison. Without it, readers including public health officials are left to guess whether differences are merely due to variations in gender and age of participants included in the study or due to the prevailing SES in their country.
Author’s response: Thank you for your suggestion. We agree and in our revised manuscript, we have presented findings from the regression models adjusted for the independent variables (sex, age, ethnicity, socioeconomic and education level).
6/30/2020 comment: concern was successfully addressed except regarding ethnicity (see below).
Author’s response: Thank you.
- In this context is it confusing why the authors would present a multilevel analysis adjusting for ethnicity. Ethnicity is a common factor causing health disparities and its adjustment can serve to hide this concerning finding. Instead, the authors should be encouraged to present an analysis whether ethnicity is a risk factor so that public resources can be directed to those who need them most. Based on the results in Table 1, it would appear that Black and "other" ethnicity is indeed a risk factor here. Again, this should be further analyzed and described.
Author’s response: Thank you for the suggestion. We have added a new table (table 4) to present the association between ethnicity and sedentary behavior adjusted for sex, age, socioeconomic and education level.
6/30/2020 comment: when analyzing for ethnicity as a predictor it is inappropriate to adjust for socioeconomic and education level variables because in many societies these are correlated with ethnicity. Regressing them out pretends all ethnicity have equal income and educational opportunities which is factually incorrect in most countries and serves to hide the impact of ethnicity on health outcomes. Therefore, ethnicity as a predictor should retain variability due to socioeconomic and education factors.
Author’s response: We thank the reviewer for the comment, but do not agree on the non-adjustment by socioeconomic factors in the analysis. We consider that the ethnic issue can be a determinant of sedentary behavior, as it is a determinant in relation to several other behaviors and health outcomes. However, the adjustment by sociodemographic variables has the objective of assessing whether ethnicity itself is an independent risk factor for the adoption of sedentary behavior. In this sense, adjusted analyses can show that regardless of income or highest academic achievement, we can know if a certain ethnicity has greater or lesser sedentary behavior when compared to another.
- From line 150 until 206, the authors are duplicating a lot of table information in the text. The redundant presentation of a lot of number in text when those numbers appear in tables is generally discouraged. Instead, authors should be encouraged to discuss important findings from the tables in general terms with particular emphasis on effect sizes and patterns, such as ethnic disparities, that casual inspection of tables may not reveal.
Author’s response: We thank the reviewer for this comment. We have now focused our results’ description on the size of the difference in clinically relevant values.
6/30/2020 comment: I would suggest making Table 7 the new Table 2 since Table 7 pertains to overall SB and old Tables 2 through 6 to domain specific SB. This would also lay the conceptual foundation for covariate adjustments in subsequent analyses.
Also, on line 128 the authors now report the ICC for ‘talking on the telephone’ is .06 which means this is not a reliable category (scores are essentially random). Therefore, this SB domain should be eliminated from all analyses.
The early part of the results section, until line 193, still contains a lot of duplicated table numbers and I would encourage the authors to edit that part of the results section to look more like lines 194ff.
Author’s response: Thank you for pointing this out.
1 – As requested, we have changed the order of the tables;
2 - We understand the suggestion to exclude these results from the analyses, but we believe that it is most appropriate to present the results of the full questionnaire (all domains of sedentary behavior), despite the ICC: 0.06 for the ‘talking on the telephone’. Thus, to assist readers, we insert a paragraph of this item in the limitations of the study. In addition, manuscripts that used the same questionnaire, presented low values of ICC (i.e., 0.04 to 0.14) for some domains of sedentary behavior.
Gardiner PA, Clark BK, Healy GN, et al. Measuring older adults' sedentary time: reliability, validity, and responsiveness. Med Sci Sports Exerc 2011, 43, 2127-2133.
Healy GN, Clark BK, Winkler EAH, et al. Measurement of adults’ sedentary time in population-based studies. Am J Prev Med 2011;41(2):216 –227.
Dempsey PC, Hadgraft NT, Winkler EAH, et al. Associations of context-specific sitting time with markers of cardiometabolic risk in australian adults. Int J Behav Nutr Phys Act. 2018;15(1):114.
3 – As requested, we have edited the first part of the results section.
- Tables 2 through 5 need clarification as to whether standardized or non-standardized regression coefficients are reported. If these are non-standardized coefficients, they should minimally be starred to indicate their statistical significance and, more importantly, their effect size.
Author’s response: We thank the reviewer for this comment. We reported unstandardized coefficients and as previously noted, we have adjusted the nomenclature in the statistics section as well as in the tables (as suggested). Also, we have now focused our results’ description on the size of the difference in clinically relevant values.
6/30/2020 comment: successfully addresses the concern.
Author’s response: Thank you.
- In line 225, the authors again claim that TV watching is of particular concern which is not accurate (see above).
Author’s response: To address your concern, we have reworded and revised this statement accordingly.
6/30/2020 comment: successfully addresses the concern.
Author’s response: Thank you.
- The discussion also suffers from the same problem as the abstract in that there is little emphasis on correlates of total SB and much of differences in particular types of SB. While there is some evidence of marginal differences in the health effects based on type of SB, there is extensive evidence that excessive SB is harmful. I would therefore encourage detailed analysis of correlates of total SB time including references to clinical benchmarking, i.e. what magnitude of difference in total SB time is expected to make a real difference regarding health outcomes.
Author’s response: As per your suggestion, we have added a new analysis (association of total sedentary time with each independent variable separated by country) presented in Table 7. In addition, we have inserted a paragraph to address and discuss the findings with respect to total sedentary behavior.
6/30/2020 comment: The discussion section still lacks a discussion that focusses clearly on correlates of the amount of SB. Again, there is little evidence SB domains impact health; there is lots of evidence overall SB amount impacts health. I would suggest clearly and cleanly discussing correlates of overall SB time first. Once that discussion is done, discussion of SB domains can be done but this should be de-emphasized and qualified since, again, there is comparatively little evidence that a person’s health is impacted by what they do while sedentary.
Author’s response: Thank you. We have strived to address your comment and include this information in the Discussion section.
- Finally, there is no discussion of the ethnic disparities evident in Table 1. Again, these should be elucidated through further analysis, such as hierarchical regression controlling for SES, and then discussed. With these changes, the manuscript can make an important contribution to the literature on SB.
Author’s response: Thank you for your suggestion. We have added a new table that shows regression findings concerning the association between ethnicity and domain-specific sedentary behavior by country.
6/30/2020 comment: Clear (or really any) discussion of ethnicity as a correlate of total SB time is lacking. The same is true for all other correlates. E.g. What factors put ethnic minorities at risk for increased SB time, e.g. residence in neighborhoods where walking or exercise is unsafe, lack of access to gyms, etc. Same is true for all other factors: Why are men at greater risk etc.
Author’s response: In line with your suggestion, we have now added information about the topic in the Discussion.
